# Differentiable Hierarchical Visual Tokenization

**Marius Aasan**[1],    **Martine Hjelkrem-Tan**[1],    **Nico Catalano**[2],
**Changkyu Choi**[3],    **Adín Ramírez Rivera**[1]

[1] University of Oslo
Deptartment of Informatics

[2] Polytechnic University of Milan
Artificial Intelligence and Robotics Lab

[3] UiT The Arctic University of Norway
Department of Physics and Technology

mariuaas@uio.no, matan@uio.no, nico.catalano@polimi.it,
changkyu.choi@uit.no, adinr@uio.no

## Abstract

Vision Transformers rely on fixed patch tokens that ignore the spatial and semantic structure of images. In this work, we introduce an end-to-end differentiable tokenizer that adapts to image content with pixel-level granularity while remaining backward-compatible with existing architectures for retrofitting pretrained models. Our method uses hierarchical model selection with information criteria to provide competitive performance in both image-level classification and dense-prediction tasks, and even supports out-of-the-box raster-to-vector conversion.

## 1 Introduction

Transformers [1] have become the de facto architecture for all but a few data modalities [2–8]. The architecture is, however, contingent on the process of *tokenization*. Tokenizers for natural language [9, 10] are designed to compress text into morphemes and semantic subwords—minimal units aligned with meaning. Yet in vision, tokenization is comprised of partitioning images into uniform square patches, ignoring semantic content and object boundaries in favor of computational convenience. This highlights a key incongruity; text tokenizers align with semantic units while patch-based vision tokenizers fragment objects without regard for their structure. Figure 1 illustrates how patch tokens lack the semantic coherence and granularity necessary for dense predictions.

Efforts to move beyond the grid paradigm include clustering or merging [11, 12] for grouping features dynamically, or deformable patches [13, 14] to improve spatial adaptivity. Recent work propose *subobject tokenizers* [15–17] to partition images into semantically coherent regions, providing gains in classification, segmentation accuracy, and interpretability. However, each method tackles only one or two facets of the larger tokenization problem, and none are fully end-to-end learnable.

An effective visual tokenizer must unify precise semantic alignment, differentiability, and adaptive granularity. Our key insight is that hierarchical pixel-level partitioning can be formulated as a multi-scale model selection problem, and can be combined with differentiable mechanisms for end-to-end learning. We propose *differentiable hierarchical tokenization* ($\partial$HT) for Vision Transformers (ViTs), emphasizing modularity [18, 19] for reuse and backward compatibility.

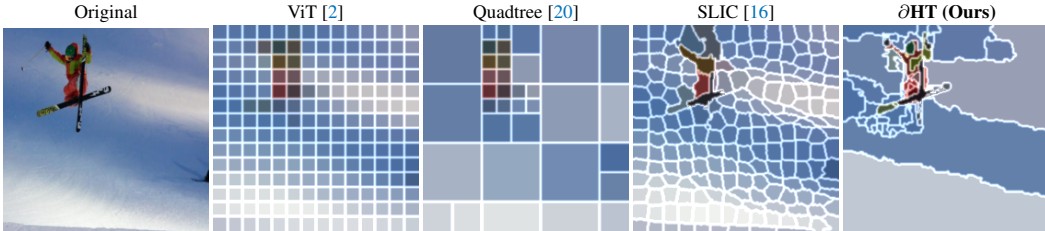

**Figure 1:** Comparing spatial granularity in visual tokenizers. $\partial$HT (right) provides an end-to-end learnable framework for multi-scale tokenization. We provide more examples in Figure E.6.

Code and model weights: https://github.com/dsb-ifi/dHT

39th Conference on Neural Information Processing Systems (NeurIPS 2025).

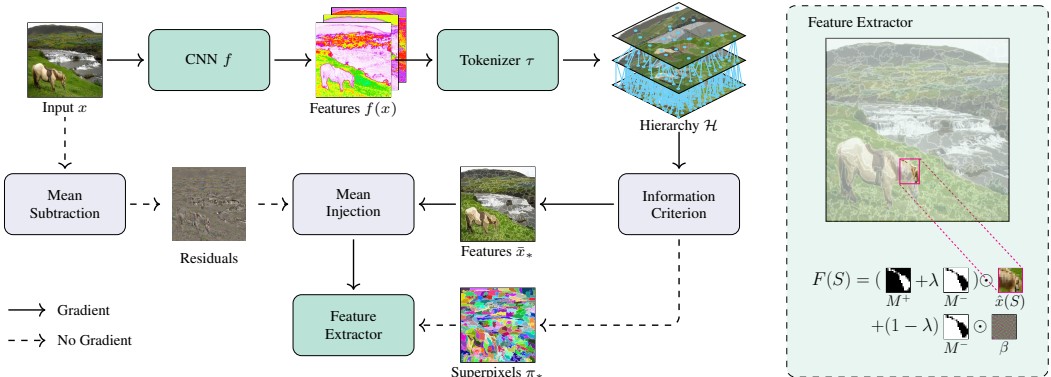

**Figure 2:** Illustration of the $\partial$HT tokenization and feature extraction pipeline. From an input image we produce a hierarchy of superpixel representations. An optimal segmentation is then selected from the hierarchy using information criteria, and features are extracted for each superpixel. Features can then be used in any ViT backbone. (Right) We also depict the feature extraction process of a superpixel $S$ where its features are mixed based on foreground, $M^+$, background, $M^-$, and shared background features, $\beta$. Details in Section 2.4.

Our contributions can be summarized as follows:

• **Learnable tokens:** A novel end-to-end differentiable tokenization method which adapts to training data, and ensures effective tokenization for classification and segmentation tasks.
• **Retrofitted tokenizers:** A flexible fine-tuning strategy to adapt pretrained ViTs for superpixel tokens with pixel-level granularity, enhancing versatility across tasks.
• **Multiscale model selection:** A lattice theoretic extension of information criteria to multilevel hierarchical partitioning for selecting the most informative image partitions.
• **Image vectorization:** A out-of-the-box method for adapting hierarchical superpixel tokenizers to raster-to-vector graphics conversion, without being specifically trained for this task.

## 1.1 Motivation

Visual tokenization means discovering a discrete set of regions from a continuous, high-dimensional image, under strict compute and memory budgets—effectively solving segmentation, compression and representation learning all at once. Unlike 1D sequences, where you have a natural ordering and can pick breakpoints by cues such as whitespace and morpheme statistics, spatial data offer no canonical order. The space of possible region shapes, sizes and connectivity explodes, and tokens must respect multi-scale structure and spatial invariances. Moreover, full integration with ViTs require *differentiable tokenization*—adaptively choosing token count, placement, and shape to adhere to semantic boundaries to extract meaningful atomistic units from a scene with end-to-end learning.

## 1.2 Related Work

Existing approaches to adaptive visual tokenization can be organized into three broad categories.

**Dynamic Grouping and Merging:** Several works [11, 20–22] propose to dynamically cluster or merge grid tokens within the transformer's layers. These approaches reduce redundancy and adapt token budgets per image, but remain tethered to the original grid primitives and do not introduce truly off-grid partitions. This prevents discovery of true subobject boundaries for semantic coherence.

**Deformable Sampling:** A different approach stems from replacing fixed patches with learned sampling with deformation and dynamic positions. DPT [13] and Deformable Attention Transformer (DAT) [14] learn per-token centers and scales via differentiable bilinear sampling, granting geometric flexibility. While fully end-to-end learnable, these are still fundamentally patch based, leaving semantic alignment and object boundaries implicit.

**Subobject and Superpixel Tokenizers:** Superpixel-based methods partition images into semantically coherent regions prior to encoding. SuiT [16] pools CNN features over SLIC superpixels; EPOC [17] combines learned boundary cues with watershed grouping to form panoptic tokens; SP-

former [23] uses hybrid cross-attention pixel-to-token clustering between attention blocks; SPiT [15] performs parallel hierarchical graph merging.

While SuiT works on a downsampled version of the input image, other methods provide true pixel-level granularity for extracting subobject tokens. These yield clear gains in classification, segmentation, and interpretability, but the non-differentiable grouping step prevents the tokenizer itself from being trained end-to-end.

## 2 Method: Differentiable Hierarchical Tokenization

Our proposed method builds on previous subobject level approaches to provide the first fully end-to-end learnable tokenizer for ViTs. We emphasize that $\partial$HT is not another ViT variant, but a fully modular tokenizer that can serve as a *plug-and-play* extension for pretrained models. Our design emphasizes precise semantic alignment with pixel-level granularity, multi-scale awareness via hierarchical pruning with information criteria, end-to-end differentiability, and modularity.

**High Level Overview:** $\partial$HT produces adaptive tokens through four stages, illustrated in Figure 2.

- *Feature projection* (Section 2.1): We embed each pixel into a learned $d$-dimensional feature space using a lightweight CNN, establishing a similarity metric for subsequent grouping.
- *Hierarchical partitioning* (Section 2.2): Starting from individual pixels, we iteratively merge similar adjacent regions to construct a complete hierarchy, capturing structure at multiple scales.
- *Optimal selection* (Section 2.3): Information criteria identify the partition that best balances model fit against complexity, eliminating manual threshold tuning.
- *Differentiable extraction* (Section 2.4): A mean-injection mechanism produces token features compatible with standard ViTs while enabling end-to-end gradient flow.

**Preliminaries:** Consider a graph $G = (V, E)$ where the vertices represent pixel positions in a grid of width $w$ and height $h$, and edges $E$ are connections of horizontally and vertically adjacent vertices. We view an image as a signal function $x \colon V \to \mathbb{R}^c$, mapping each vertex $v \in V$ to a $c$-channel pixel feature. A *connected partition* of $V$ is a set $\pi = \{S : S \subseteq V\}$ which satisfies:

(i) *Non-overlapping*: For any pair $S, S' \in \pi$, their intersection is empty, i.e., $S \cap S' = \emptyset$.
(ii) *Covering*: The union of all $S \in \pi$ is the full vertex set, i.e., $\bigcup_{S \in \pi} S = V$.
(iii) *Connected*: For any vertices $u, v \in S$, there exists a path of adjacent vertices in $S$ starting at $u$ and ending at $v$, where each pair of consecutive vertices is connected by an edge in $E$.

Let $\Pi(V)$ be the space of all partitions of $V$ and consider two partitions $\pi_1, \pi_2 \in \Pi(V)$. If for every region $S_1 \in \pi_1$ there exists a region $S_2 \in \pi_2$ such that $S_1 \subseteq S_2$, we say that the two partitions form a *hierarchy* $\mathcal{H} = (\pi_1, \pi_2)$ ordered by refinement.

### 2.1 Subobject Feature Projection

We use a lightweight CNN encoder $f \colon \mathbb{R}^c \to \mathbb{R}^d$ to embed each pixel to an initial feature space $f_0(v) = f(x(v))$. We design $f$ with a residual branch that applies a $1 \times 1$ convolution to lift the $c$-channel input to $d$ dimensions, and a main branch that downsamples via two successive convolutions ($3 \times 3$ or $2 \times 2$ strided) followed by bilinear upsampling.

### 2.2 Hierarchical Vertex Merging

We expand on existing methods [15] to construct hierarchical partitions by iterative vertex merging. On a high level, the procedure pairs vertices with their most similar neighbor and merges them via parallel connected components, ensuring that each region is a connected superpixel.

More formally, given a positive semi-definite kernel $\kappa \colon V \times V \to \mathbb{R}_{\geq 0}$, we define the maximally similar vertex $v_{\max} = \arg\max_{u \in \mathfrak{N}(v)} \kappa(u, v)$ such that

$$E_{\max} = \{(v, v_{\max}) : v \in V\} \subset E, \tag{1}$$

pairs each vertex with its most similar neighbor within the neighboring vertices $\mathfrak{N}(v)$ defined by $E$.

Let $\mathcal{C}_1 \colon V \to V_1$ denote a mapping of vertices to the connected components of the *spanning subgraph*[1] $G[E_{\max}]$. Then, $\mathcal{C}_1(v)$ denotes a connected superpixel region $S \in V_1$ such that vertices within the same component of $G[E_{\max}]$ are assigned to $S$. In particular, $S$ contains $v$, $v_{\max}$, and potentially other vertices based on similarity. A natural choice for updating the vertex features for $V_1$ is to simply take the average feature values in each component

$$\bar{f}_1(v) = \frac{1}{|S|} \sum_{u \in S} f_0(u), \quad S = \mathcal{C}_1(v). \tag{2}$$

For a given level $t$, the mapping $\mathcal{C}_t$ induces a graph contraction $G_{t-1} \mapsto G_t$. Each vertex $v \in V_{t-1}$ is assigned to a connected component $\mathcal{C}_t(v) = S_t \in \pi_t$, which yields a hierarchy of partitions $\mathcal{H} = (\pi_0, \ldots, \pi_T)$. Importantly, each superpixel $S_t$ at any level $t$ in the hierarchy is always grounded in the original pixel positions in $V$. We formalize the construction in Appendix A.2.

## 2.3 Hierarchical Pruning via Information Criteria

Having constructed a full hierarchy, we now look to select an optimal partition. The construction of $\mathcal{H}$ is designed to tackle two objectives; multi-scale adaptability and redundancy management. We compute the hierarchy up to a singleton such that all regions are connected up to a root region representing the full image. This allows us to frame the search for an optimal partition $\pi_* \in \Pi(V)$ as a *model selection problem* using information criteria (IC) [24], removing the need for threshold calibration or gating [20, 25, 26]. Recall that an IC has the generalized form

$$\mathrm{IC}(\theta) = -2 \log L(\theta) + g(\mathrm{df}_\theta), \tag{3}$$

where $L(\theta)$ is the likelihood of the data under the parameter $\theta$, and $g(\mathrm{df}_\theta)$ is a penalty function based on the statistical degrees of freedom $\mathrm{df}_\theta$ that discourages overly complex models. A choice of IC typically determines the form of $g$. Our goal is then to derive an estimate for $\mathrm{IC}(\pi_*)$.

**Likelihood**: A superpixel representation $v \in S \in \pi_*$ can be considered a *piecewise constant model* of the image for which each region $S$ is assigned a constant value $\mu_S$. We model each pixel feature $f_0$ as i.i.d. samples from a Gaussian distribution, i.e., $f_0 \mid S \sim \mathcal{N}(\mu_S, \Sigma_S)$. Under this assumption, the log-likelihood $\log L(\pi_*)$ simplifies to terms only involving $\Sigma_S$. This aligns with variance reduction criteria [25], but augmented with additional parsimony constraints. See Appendix B.1 for details.

**Degrees of freedom:** Traditionally, degrees of freedom $\mathrm{df}_\theta$ correspond to the number of parameters estimated in a model. For each superpixel $S$, $\mu_S$ is a parameter in a piecewise constant model of image $x$, making $\mathrm{df}_\theta$ proportional to the total number of regions in the optimal partition $\pi_*$. However, this cannot be determined exactly without search of all possible combinations in $\mathcal{H}$. Instead, we leverage the lattice structure [27] of partition space $\Pi(V)$, and find that $\mathrm{df}_\theta$ are inversely proportional to the total number of connected edges within each $S$. This gives a proportional estimate

$$\mathrm{df}_{\pi_*} = \sum_{S \in \pi_*} \mathrm{df}_S \propto \sum_{S \in \pi_*} \mathrm{Vol}_G(S)^{-1}, \tag{4}$$

where $\mathrm{Vol}_G(S)$ represents the total number of edges $(u, v) \in E$ such that $u, v \in S$. This formulation effectively penalizes partitions that consist of smaller regions (with fewer internal edges), encouraging larger, more informative regions while still capturing essential structural information. We provide a formal derivation of this result in Appendix B.2.

In turn, $\partial$HT prunes a full hierarchy $\mathcal{H}$ according to the selected IC, balancing model fit and complexity to select optimal partitions. We explore the effects of different information criteria on our method's performance in Section 3.3.

## 2.4 Differentiable Token Embeddings

There are two main approaches to feature extraction with superpixel tokenization. The most common approach is to perform aggregation of regions through a separate encoder [16, 17]. However, this prevents drop-in replacement in a ViT model, making retrofitting tokenizers to pre-trained models non-trivial. In contrast, SPiT [15] generalizes the ViT feature extraction process to irregular regions. Each region's bounding box is interpolated to a fixed patch size while masking out the background

---

[1]See Definition A.2 for details.

features of the surrounding vertices. This is backward compatible with standard ViTs, but suffers from being inherently non-differentiable. $\partial$HT makes generalized ViT features fully differentiable by introducing weighted aggregation and a novel *mean-injection trick* reminiscent of straight-through estimation [28, 29].

At each level $t$, we collect learning signals from the contraction process by computing features via

$$f_{t+1}(v) = \frac{|S_t|}{|S_{t+1}|} \sum_{u \in S_{t+1}} \kappa(u, u_{\max}) \cdot f_t(u), \tag{5}$$

where $|S_t|$ and $|S_{t+1}|$ denote the number of pixels in the respective superpixel regions at two consecutive levels in the hierarchy. This update is a *weighted variant* of (2) with each vertex's contribution weighted by the kernel score of their most similar neighbor at step $t$.

Additionally, let $f_*(v)$ denote the feature of vertex $v \in V$ under some optimal partition $\pi_*$. To extract features from interpolated regions of the original image $x$, the idea is to inject the pruned vertex features into the original image channels without altering its local texture properties. For each pixel $v \in S \in \pi_*$, we adjust the original pixel feature $x(v)$ by computing

$$\hat{x}(v) = x(v) + W f_*(v) - \bar{x}_*(v), \tag{6}$$

where $W \colon \mathbb{R}^d \to \mathbb{R}^c$ is a learnable linear mapping. This effectively replaces the mean of each superpixel region $\bar{x}_*(v) = \frac{1}{|S|} \sum_{u \in S} x(u)$ with a learnable estimate from the tokenization process, enabling full gradient flow from tokenization to the ViT backbone.

**Positional Embedding:** Previous work has shown that positional embedding computed as a linear combination of kernelized joint-histogram positions for each region generalize standard learnable positional embeddings [15]. We extend previous work by allowing for higher resolution, which provides more fine grained detail to token embeddings, and ablate the effect in Table 8.

**Modularity and Retrofitting:** Modularity is central to developing complex systems [18, 30], and allows architectures to be broken down into reusable components. Just as transfer learning lets you fine-tune pretrained task heads, a modular tokenizer enables you to *transfer pretrained models across distinct tokenization schemes* without retraining from scratch. To apply this principle to a pre-trained patch-based ViT, we initialize $\partial$HT as follows. From (6), we observe that having $W f_*(v) = \bar{x}_*(v)$ results in $\hat{x} = x$; a perfect reconstruction of the original image. We find that pretraining the encoder $f$ and the linear mapping $W$ jointly with

$$\mathcal{L}_{\text{rec}} = \frac{1}{|V|} \sum_{v \in V} \left\| x(v) - W f_*(v) \right\|_2^2 \tag{7}$$

provides maximally aligned features for fine-tuning a ViT backbone with $\partial$HT in place of the canonical tokenizer, allowing for fully differentiable tokenizer retrofitting.

**Background Masking:** During our experiments on retrofitting, we observe that masking out background features leads to sparsity when highly irregular regions are interpolated to a fixed size. Intuitively, masking leads to a minor domain shift in token embeddings, since the original features are dense within a patch and masking naturally leads to sparser representations. This can result in slower convergence and loss in performance.

In response, we introduce dynamic adaption of background masks during training. Let $q$ denote the feature patch size, and let $\lambda \in [0, 1]$, $\beta \in \mathbb{R}^{c \times q \times q}$ be learnable parameters in a feature extractor. Let $M_S^+ \in \{0, 1\}^{q \times q}$ be the interpolated foreground mask of a superpixel $S$, with $M_S^- = 1 - M_S^+$ denoting the background mask. We extract token features $F(S) \in \mathbb{R}^{c \times q \times q}$ via

$$F(S) = (M_S^+ + \lambda M_S^-) \odot \hat{x}(S) + (1 - \lambda) M_S^- \odot \beta, \tag{8}$$

where $\odot$ is element-wise product. This tweak allows the model to blend the foreground and background elements within each region through $\lambda$. The parameter $\beta$ serves as a shared background feature, which mitigates sparsity by preventing zeroing out background regions completely. An illustration is provided in the right section of Fig. 2.

We ablate the effect of masking, encoder $f$, choice of kernel $\kappa$ and information criteria IC, as well as other hyperparameters and architectural specifics of $\partial$HT in Tables 6 and 7.

**Table 1:** Classification top-1 and kNN accuracies ($224 \times 224$). The top six rows (above the midline) show results for the baseline models and retrofitted (RF) $\partial$HT counterparts. The bottom four rows show results for models trained from scratch. Models are trained exclusively on ImageNet1k (IN).

| Token. | Pretraining | Model | IN-Val [31] Acc@1 | IN-Val kNN | IN-ReaL [32] Acc@1 | IN-ReaL kNN | IN-v2 [33] Acc@1 | IN-v2 kNN | Caltech†[34] Acc@1 | Caltech kNN | CUB†[35] Acc@1 | CUB kNN | Cars†[36] Acc@1 | Cars kNN |
|---|---|---|---|---|---|---|---|---|---|---|---|---|---|---|
| Patch | DEiT3 [37] | ViT-S16 | 80.4 | 80.7 | 86.1 | 86.9 | 69.7 | 58.3 | 87.0 | 77.3 | 54.6 | 38.4 | 24.9 | 6.5 |
| | DEiT3 [37] | ViT-B16 | 82.6 | **83.1** | 87.7 | **88.2** | 72.6 | **63.0** | 90.3 | 78.0 | 72.1 | 49.0 | 35.1 | 10.5 |
| | TIMM [38] | ViT-B32 | 70.2 | 69.5 | 77.0 | 76.9 | 54.4 | 55.9 | 87.1 | 73.6 | 74.8 | 38.0 | 21.0 | 4.0 |
| $\partial$HT | DEiT3-RF | ViT-S16 | 80.1 | 78.4 | 84.9 | 83.7 | 68.2 | 58.1 | 88.1 | 77.4 | 63.1 | 46.4 | 31.1 | 8.3 |
| | DEiT3-RF | ViT-B16 | **83.9** | 82.9 | **88.3** | 87.7 | **73.2** | **63.0** | **91.2** | **79.1** | 74.4 | **52.5** | 34.7 | 9.2 |
| | TIMM-RF | ViT-B32‡ | 83.1 | 81.6 | 87.3 | 87.4 | 72.2 | 62.4 | 90.2 | 78.3 | 72.8 | 50.1 | **39.8** | **12.9** |
| Patch | From Scratch | ViT-S16 | 79.9 | 77.5 | 83.9 | 84.9 | 68.1 | 57.4 | 86.1 | 76.4 | 57.2 | 49.7 | 21.4 | 6.3 |
| | | ViT-B16 | 81.9 | **80.4** | 86.7 | 87.3 | 71.1 | **62.1** | 88.4 | 78.0 | **73.3** | **52.0** | 31.3 | 8.4 |
| $\partial$HT | | ViT-S16 | 80.0 | 77.6 | 84.1 | 85.9 | 69.2 | 57.2 | 87.5 | 76.3 | 69.6 | 53.8 | 23.2 | 7.9 |
| | | ViT-B16 | **83.2** | 80.3 | **87.4** | **88.2** | **72.8** | **62.1** | **89.3** | **78.1** | 73.2 | 51.6 | **33.0** | **9.3** |

†Frozen backbone and linear probing.
‡Note that adaptive tokenization results in higher numbers of tokens compared to baseline.

# 3 Experiments and Results

We design our experiments to investigate the representation capabilities of our method's extracted tokens in multiple settings; including end-to-end learning with classification on ImageNet1k [31], transfer learning as a drop-in tokenizer replacement for pretrained ViTs (cf. Section 3.1), and demonstrating decoder-free segmentation models with learnable tokenization (cf. Section 3.2). Moreover, $\partial$HT can also be evaluated on learnable image vectorization, and we compare our method to learnable image vectorization models (cf. Section 3.2). Training setup is detailed in Appendix D.

## 3.1 Image Level Predictions

We focus on transformer baselines trained exclusively on ImageNet1k [31], and validate on various downstream tasks [32–36]. In addition to reporting top-1 accuracy scores, we perform a kNN evaluation to assess the quality of the representation space. kNN scores are computed by taking the max score over $k \in \{10, 20, 50, 100, 150, 200\}$ [39].

**Retrofitting:** We evaluate the effect of retrofitting $\partial$HT to pretrained models [37], including the less common B32 capacity [38] for completeness. We align the $\partial$HT tokenizer by pretraining using (7). Then, we finetune pretrained models to replace the canonical tokenizer in the ViT that will be retrofitted with $\partial$HT tokenizer. All results are uniformly re-evaluated for fair comparison, explaining minor differences from previously reported results [37].

Table 1 (top) shows that tokenizer retrofitting has a generally positive effect on linear evaluation models with base capacity, while maintaining competitive performance for small capacity models. Interestingly, our kNN results for ImageNet1k indicate that DEiT3 [37] models generally perform better with kNN than the linear evaluation, which is surprising as one typically expects the opposite. Contrarily, our retrofitted counterparts yield an expected result; the linear head produces better results than kNN. On the other hand, our retrofitted models are better aligned with both linear probing and kNN on Caltech256 [34] and CUB200 [35], indicating that $\partial$HT provides token representations that are useful for generalizing beyond the training data.

In general, we observe that retrofitting models with $\partial$HT enables models to adapt tokens to individual images, maintaining or slightly improving overall classification performance compared to baselines.

**Training from Scratch:** We compare transformers using $\partial$HT to standard ViT patch tokenization [2] by training models from scratch under the same training regime, such that training is *guaranteed to be equivalent*. This eliminates confounding factors such as hardware and minor implementation differences. Contrary to the retrofitted setup, models trained from scratch use local gradient features, which have been shown to improve performance for both canonical and superpixel tokenization [15] with minimal computational overhead (+0.07 GFLOPs, +256 pa-

**Table 2:** Comparison of ViT-S16 models trained from scratch on ImageNet with different tokenizers. *Mod.* denotes commensurability with ViTs, while *Diff.* denotes end-to-end differentiability.

| Method | Mod. | Diff. | Acc@1 Small | Acc@1 Base |
|---|---|---|---|---|
| Patch / DEiT [40] | ✓ | | 79.9 | 81.8 |
| Patch / DEiT3 [37] | ✓ | | 80.4 | 82.6 |
| SPFormer [12] | | ✓ | **81.7** | 82.7 |
| SPiT [15] | ✓ | | 75.0 | 80.4 |
| SuiT [16] | | | 80.9 | 82.1 |
| $\partial$HT | ✓ | ✓ | 80.0 | **83.2** |

**Table 3:** Single Scale Semantic Segmentation mIoU results on ADE20k [41] and COCO-Stuff164k [42].

| Dataset | Backbone | Method | Size ($\downarrow$) | mIoU ($\uparrow$) |
|---|---|---|---|---|
| ADE20k | RN50 [43] | UperNet [44] | 640 | 40.1 |
| | Swin-B [45] | Mask2Former [46] | 640 | 52.4 |
| | MiT-B5 [23] | Segformer [23] | 640 | 51.0 |
| | SegVit-B [47] | ATM [47] | 512 | 51.3 |
| | BiFormer-S [48] | UperNet [44] | 640 | 49.8 |
| | BiFormer-B [48] | UperNet [44] | 640 | 51.0 |
| | ConvNext-L | SP-Transformer [49] | 640 | 43.7 |
| | SPFormer-S [12] | SPFormer | 640 | 46.5 |
| | MAE-B [50] | UperNet [44] | 640 | 47.1 |
| | MAE-H [50] | Linear | 640 | 33.3 |
| | DINOv2-S [51] | Linear | 640 | 44.3 |
| | DINOv2-B [51] | Linear | 640 | 47.3 |
| | DINOv2-B [51] | Linear | 640 | 47.3 |
| | $\partial$HT-ViT-S | MLP | 512 | 47.1 |
| | $\partial$HT-ViT-B | MLP | 512 | **53.2** |
| COCO164k | MiT-B5 | Segformer [23] | 640 | 46.7 |
| | MiT-B5 | Lawin [52] | 640 | 47.5 |
| | Swin-B | UperNet-RR [53] | 640 | 48.2 |
| | ViT-L | Segmenter [54] | 512 | 48.4 |
| | $\partial$HT-ViT-B | MLP | 512 | **48.9** |

**Table 4:** Zero-shot segmentation results on salient detection datasets using token-cut. $\partial$HT outperforms existing approaches, including other adaptive tokenization frameworks.

| | ECSSD [55] | | | DUTS [56] | | | DUT-OMRON [57] | | |
|---|---|---|---|---|---|---|---|---|---|
| Backbone | $F_{max}$ | IoU | Acc@1 | $F_{max}$ | IoU | Acc@1 | $F_{max}$ | IoU | Acc@1 |
| DINO-B [58] | 80.3 | 71.2 | 91.8 | 67.2 | 57.6 | 90.3 | 60.0 | 53.3 | 88.0 |
| DINO-B[†] [58] | 87.4 | 77.2 | 93.4 | 75.5 | 62.4 | 91.4 | 69.7 | **61.8** | 89.7 |
| SPiT-B [15] | 90.3 | 77.3 | 93.4 | 77.1 | 63.9 | 89.4 | 71.1 | 56.4 | 86.8 |
| SuiT-B [16] | 87.0 | **80.5** | 93.8 | 68.0 | 60.0 | 88.8 | 63.4 | 57.5 | 87.2 |
| $\partial$HT-B | **92.4** | 79.9 | **94.2** | **77.9** | **64.4** | **90.5** | **71.9** | 58.7 | **89.8** |

[†]Applies post-processing via bilateral upsampling.

**Table 5:** Results on raster-to-vector conversion. We compare with a off-the-shelf baseline (Adobe) and a differentiable model specifically trained for the task.

| | | DiffVG-5 | | |
|---|---|---|---|---|
| Method | Zero-shot | MSE($\downarrow$) | PSNR($\uparrow$) | SSIM($\uparrow$) |
| Adobe | | 0.00712 | 21.84 | 0.7318 |
| DiffVG-MC [59] | | 0.00316 | 25.54 | 0.8287 |
| DiffVG-AP [59] | | 0.00265 | 26.22 | 0.8494 |
| $\partial$HT | ✓ | **0.00178** | **27.50** | **0.8541** |

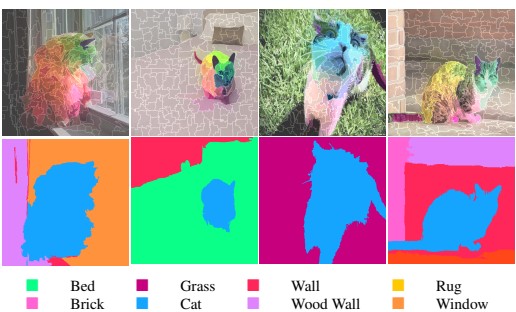

| | | | |
|---|---|---|---|
| 🟩 Bed | 🟪 Grass | 🟥 Wall | 🟨 Rug |
| 🟪 Brick | 🟦 Cat | 🟪 Wood Wall | 🟧 Window |

**Figure 3:** $\partial$HT tokenized images with feature correspondences (top) and semantic segmentation (bottom).

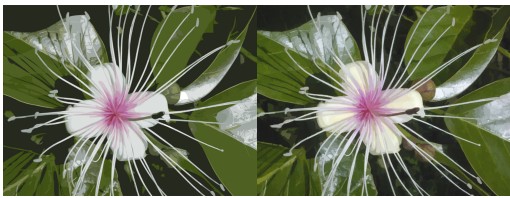

**Figure 4:** Comparison of image vectorization with DiffVG [59] (left) and $\partial$HT (right).

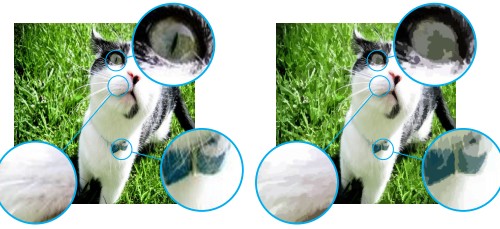

**Figure 5:** Image vectorization from $\partial$HT token extraction with zooms to show the finer details, comparing the original image (left) and vectorized image (right).

rameters). Our results show that superpixel tokenization generally provides stronger classification results than canonical tokenization with patches, cf. Table 1 (bottom) in a strict apples-to-apples comparison. We also perform a comparison with existing superpixel-based tokenization approaches in Table 2. Our results show that $\partial$HT for ViT-B outperforms other methods while preserving modular compatibility with ViT architectures.

## 3.2 Dense Predictions

We assess our method on dense tasks by evaluating the fully trained $\partial$HT models for semantic segmentation, zero-shot segmentation performance on selected benchmarks, as well as the non-standard task of *image autovectorization*—converting rasterized images to vector images.

**Semantic Segmentation:** We fine tune our fully trained $\partial$HT models on semantic segmentation baselines [41, 42], without a dense decoder for upscaling. Instead, each superpixel is individually classified as a single segment using a simple MLP head for each token. The results in Table 3 show that $\partial$HT provides raw token embeddings that are well suited for image segmentation. A qualitative analysis with examples can be found in Appendix E, with extended results in Appendix C.

**Zero-Shot Segmentation:** Table 4 shows results for fully trained $\partial$HT models using the Token-Cut [58] method over three selected salient segmentation tasks [55–57]. The results demonstrate that $\partial$HT provides strong results in zero-shot salient segmentation, and shows that tokens can be used out-of-the-box with no post-processing or specialized training required.

**Table 6:** Tokenizer hyperparameters evaluated on reconstruction based pretraining on ImageNet.

| Metric | Dimension $d$ | | | | | | Kernel function $\kappa$ | | | Information criterion | | | |
|---|---|---|---|---|---|---|---|---|---|---|---|---|---|
| | 3 | 6 | **8** | 16 | 24 | 32 | Cosine | Tanimoto | **Gaussian** | AIC | **AICC** | BIC | CIC |
| MSE ($\downarrow$) | 0.09 | 0.08 | **0.06** | 0.07 | **0.06** | **0.06** | 0.10 | 0.08 | **0.06** | 0.08 | **0.06** | 0.07 | 0.11 |
| SSIM [60] ($\uparrow$) | 0.56 | 0.55 | **0.60** | 0.58 | 0.57 | 0.58 | 0.54 | 0.55 | **0.60** | 0.52 | **0.60** | 0.57 | 0.49 |

**Table 7:** Architectural ablations for $\partial$HT components. Evaluated with ViT-S16 over ImageNet.

| | Acc@1 | |
|---|---|---|
| Ablation | DEiT3-RF | From Scratch |
| $\partial$HT baseline | 80.1 | 80.0 |
| $-$ CNN | 76.9 ($\downarrow$ 3.2) | 75.5 ($\downarrow$ 4.5) |
| $-$ mask blending | 71.8 ($\downarrow$ 8.3) | 78.9 ($\downarrow$ 1.1) |
| $-$ $\beta$-mask features | 79.7 ($\downarrow$ 0.4) | 79.1 ($\downarrow$ 0.9) |
| $-$ kernel aggr. | 80.0 ($\downarrow$ 0.1) | 79.7 ($\downarrow$ 0.3) |
| $-$ IC pruning[†] | 80.1 ($=$ 0.0) | 79.9 ($\downarrow$ 0.1) |

†: Removing model selection generally increases tokens per image and computational cost.

**Table 8:** Ablation on sizes for kernel positional embeddings, Acc@1 on ImageNet and mIoU on ADE20k.

| Capacity | Resolution / GFLOPs | Lin@1 | mIoU |
|---|---|---|---|
| S16 | $16 \times 16$ / 0.07 | 78.4 | 36.4 |
| | $24 \times 24$ / 0.15 | **80.0** | 43.1 |
| | $32 \times 32$ / 0.26 | 80.0 | 47.2 |
| | $48 \times 48$ / 0.59 | 80.1 | **47.3** |
| B16 | $16 \times 16$ / 0.07 | 81.3 | 40.6 |
| | $24 \times 24$ / 0.15 | **83.3** | 46.9 |
| | $32 \times 32$ / 0.26 | 83.4 | 50.4 |
| | $48 \times 48$ / 0.59 | 83.5 | **53.5** |

**Autovectorization and Image Tracing:** $\partial$HT provides out-of-the-box superpixel partitions that are able to represent images with very high levels of detail. By converting these regions into vectorized representations, our pretrained models can serve as a high quality raster-to-vector graphics pipeline. We extract the optimal superpixel partition, and convert each region into a vectorized path using potrace [61]. Since previous work on learnable image vectorization [59, 62] provide few quantitative baselines, we compare our method to the five examples provided by Li et al. [59] in Table 5. This shows that $\partial$HT yields high-quality raster-to-vector conversion, illustrated in Figures 4, 5 and E.3.

## 3.3 Ablations and Hyperparameters

In our ablative study, we by measuring the effect of architectural mechanisms in $\partial$HT-S for both training paradigms. For ablating $f$ as a CNN, we note that $f$ is the foundational source of gradients from the image, so it cannot simply be dropped. We therefore replace the CNN with a linear projection (i.e., a $1 \times 1$ convolution) over the input channels.

Table 7 shows that each component contributes to a drop in accuracy compared to the baseline, with the exception of model selection. This can be attributed to the fact that model selection represents a constraint on deduplication. When such constraints are removed, the number of tokens increases during training, and the model ends up using much more tokens without increase in performance.

**Tokenizer Hyperparameters:** We ablate the effect of tokenizer hyperparameters by evaluating image reconstruction quality after tokenization, using mean squared error (MSE) and structural similarity index measure (SSIM) [60]. We focus on the reconstruction of the tokens (which only requires training the tokenizer) instead of their predictive properties (which requires training the full ViT) due to our computational limitations. The ablations compare the cosine, Tanimoto [63], and Gaussian kernels, as well as the Akaike, corrected Akaike, Bayesian, and correlation information criteria (AIC, AICC, BIC, and CIC, respectively) [24].

The results in Table 6 show that a Gaussian kernel with $d = 8$ and AICC produces the best scores. Moreover, we evaluate the effect of different resolutions for the positional embeddings in Table 8. These results indicate that increasing the resolution of positional embeddings generally improve results. However, the effect saturates slightly at $24 \times 24$ for classification, and at $48 \times 48$ for segmentation. Given that increasing the resolution adds to the computational complexity (GFLOPs), we use these resolutions for our final models.

**Scale Invariance:** A central feature of $\partial$HT is that the model can select a subset of tokens that is most informative to represent each individual image. As a result, the number of tokens differs from image to image, adapting to variations in information and scale. To evaluate the effect of this behavior, we perform an experiment where we compare results over various image scales while keeping the number of tokens equivalent between models. We do this by performing an extra step of merging after the model selection step, merging regions with high similarity to produce similar numbers of tokens for each model. This test is performed comparing the retrofitted models to the baselines.

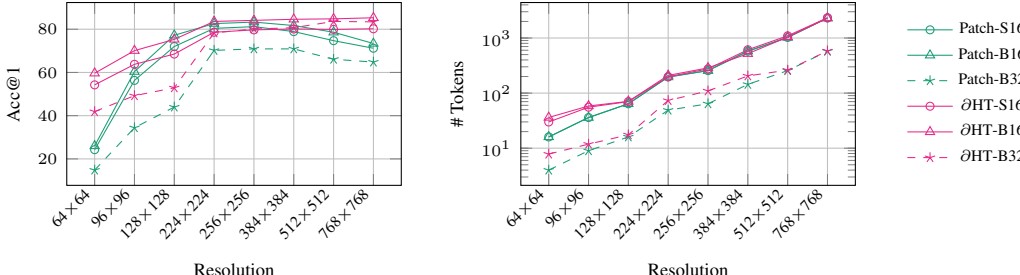

**Figure 6:** Scale invariance and token granularity for retrofitted models over ImageNet [31] with extended low-resolution evaluations (64–768px). The additional points highlight how both patch and adaptive tokenizations degrade gracefully with coarser sampling, showing stronger invariance for $\partial$HT.

Our results in Figure 6 show that $\partial$HT scales considerably better to higher resolutions by adapting tokens to image content, notably without modification of the resolution of positional embeddings.

In very low resolution settings (e.g. CIFAR's $32 \times 32$), canonical ViT tokenization may outperform $\partial$HT. In these settings, a few of pixels can end up covering an entire region and $\partial$HT may yield block-like regions similar to a square-patch tokenizer, as there are no clear inter-pixel edges to delineate. The uniform grid of patches aligns well with the reduced information content, allowing the standard tokenizer to perform adequately. $\partial$HT's adaptive token selection offers less advantage in this scenario as there is less fine-grained information to exploit. However, we note that at moderately low resolutions, $\partial$HT generally performs quite well out-of-the-box compared to patch-based ViTs.

At higher resolutions, images contain more detailed and fine-grained features, and $\partial$HT's ability to select and adapt tokens based on the most informative regions becomes highly beneficial. It can effectively capture critical details without being constrained by a fixed grid, leading to improved performance over the canonical tokenizer. $\partial$HT achieves this advantage without modifying the positional embeddings, demonstrating its inherent scalability to higher resolutions.

## 4 Discussion and Conclusion

We propose $\partial$HT, a differentiable tokenizer with pixel-level granularity that uses information criteria to dynamically select optimal partitions from hierarchical representations. Our experiments demonstrate that $\partial$HT achieves strong performance on both image-level classification and dense prediction tasks while maintaining modularity when retrofitting pretrained models.

Our work establishes tokenization as an adaptive, learnable component in ViT architectures. As models and datasets scale, this modularity becomes increasingly valuable for adapting representations to specific tasks and domains. Given the broad applicability of superpixels in vision modeling [64–67], integrating adaptive tokenization could unlock performance gains across various applications, from medical imaging to video understanding where redundancy management is critical.

### 4.1 Limitations

While $\partial$HT shows promise, it should still be taken as an early exploration of fully adaptive tokenization. Our hierarchical pruning with information criteria effectively manages redundancy but relies on modeling assumptions detailed in Appendix B. The computational overhead of superpixel tokenization currently results in lower throughput than canonical ViTs, though this gap narrows at higher resolutions where adaptive tokenization provides greater benefits. At very low resolutions, reduced semantic information limits adaptive partitioning benefits, though the method adapts well to practical image sizes.

**Table 9:** Computational efficiency of $\partial$HT with $24 \times 24$ pos. embeddings. Throughput measured on ImageNet over $8 \times$MI250x.

| Cap. | Res. | Tokenizer | Params | Tok./s | Tok./im. |
|------|------|-----------|--------|--------|----------|
| S16 | 224 | Patch | 22.1M | 192.3k | 197 |
| S16 | 224 | $\partial$HT | 22.3M | 94.2k | 240 |
| B16 | 224 | Patch | 86.6M | 89.7k | 197 |
| B16 | 224 | $\partial$HT | 86.9M | 68.2k | 246 |
| S16 | 384 | Patch | 22.1M | 112.4k | 577 |
| S16 | 384 | $\partial$HT | 22.3M | 58.1k | 494 |
| B16 | 384 | Patch | 86.6M | 61.6k | 577 |
| B16 | 384 | $\partial$HT | 86.9M | 48.9k | 432 |

Our feature extraction maintains backward compatibility with standard ViTs, but may not be optimally designed for irregular superpixel regions. Developing specialized feature extraction methods while preserving differentiability represents an important research direction.

## 4.2 Further Work

As mentioned in the limitations, architectural design choices for edge contraction and feature aggregation are promising avenues for optimal design of adaptive tokenization in vision transformers. In this work, we gather implicit learning signals weighted by the similarity kernel in (5), but extending this to explicit learnable aggregation with graph neural networks could provide more expressive modeling. However, such extensions should preserve symmetry and positive semi-definiteness to ensure consistency and well defined edge contractions in the hierarchy (Appendix A.2).

Extending differentiable tokenization to self-supervised learning represents a natural extension. This was previously explored by Lew et al. [16], which trains a DINO variant with strong results. In self-supervised settings, masked image modeling (MIM) paradigms have potential for synergy with differentiable tokenization mechanisms, providing more direct learning signals via invariants such as translation, scale, and rotation.

Vision-language models represent another promising research direction, where alignment with language could help inform the edge contraction process. Adaptive tokenization can be beneficial for document-focused tasks where fixed patches poorly align with heterogeneous text and layout structures. Preliminary investigations suggest $\partial$HT could address key limitations in current docVLM approaches by generating tokens that better capture semantic boundaries on a per-sample basis.

Additionally, video transformers present a compelling application domain. The quadratic attention complexity makes redundancy management crucial, and spatiotemporal superpixel tokenization could significantly improve efficiency while preserving semantic coherence across frames. More broadly, our findings suggest that for tasks where high dimensionality is a bottleneck, better adaptive tokenization such as $\partial$HT can provide tractable dimensionality reduction by exploiting inherent local redundancies. Such redundancies cannot be meaningfully exploited if tokens are invariant to image content, such as in the case of square patches.

### Acknowledgments

This work was funded by RCN (the Research Council of Norway) through Visual Intelligence, Centre for Research-based Innovation (309439), and in part by the RCN–NRF (National Research Foundation of Korea) joint project AURoRA (359216, RS-2025-03522980). We acknowledge Sigma2 (Project NN8104K) for access to the LUMI supercomputer, owned by the EuroHPC Joint Undertaking, hosted by CSC (Finland) and the LUMI consortium through Sigma2, Norway. We acknowledge ISCRA for access to the LEONARDO supercomputer, owned by the EuroHPC Joint Undertaking hosted by CINECA, Italy.

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

# A Theoretical Results

In this section, we provide a formal construction for monotonic edge coloring with respect to graph connectedness, and show that this construction yields a hierarchical partitioning of $V$ in Proposition A.13. We leverage this result to define a hierarchical graph partitioning in Definition A.14.

## A.1 Notation and Preliminaries

| | | | |
|---|---|---|---|
| $[\![n]\!]$ | Discrete sequence $(0, \ldots, n)$. | $\pi$ | Partition in $\Pi(V)$. |
| $2^X$ | Power set of set $X$. | $\mathcal{H}$ | Hierarchical partition $(\pi_t : \pi_t \sqsubseteq \pi_{t+1})_{t=1}^L$. |
| $G$ | An undirected graph $G = (V, E)$. | $G_t$ | Graph $G_t = (V_t, E_t)$ in hier. graph seq. |
| $x$ | $c$-channel image $x : V \to \mathbb{R}^c$. | IC | An information criteria. |
| $\mathfrak{N}(v)$ | Neigborhood of $v \in G$. | $L$ | Likelihood function. |
| $\mathcal{C}(v)$ | Connected components of $v \in G$. | df | Degrees of freedom. |
| $G[F]$ | Spanning subgraph of $G$ under $F \subseteq E$. | $\mathrm{Vol}_G(X)$ | Number of $(u, v) \in E$ s.t. $u, v \in X$. |
| $\Pi(V)$ | Set of partitions over $V$. | $\kappa(u, v)$ | Pos. def. kernel $\kappa : \mathbb{R}^d \times \mathbb{R}^d \to \mathbb{R}_{\geq 0}$. |

**Definition A.1** (Neighborhood). Let $G = (V, E)$ be a graph. The *neighborhood* of a vertex $v \in V$ is defined by $\mathfrak{N}(v) = \{u : \{u, v\} \in E\}$.

**Definition A.2** (Subgraphs). Let $G = (V, E)$ be a graph, and let $S \subseteq V$. Let $G[S] = (S, E[S])$ be a graph such that $E[S] = \{\{u, v\} \in E : u, v \in S\}$. Then $G[S]$ is a *subgraph induced by* $S$. Symetrically, for $F \subseteq E$, a subgraph $G[F] = (V, F)$ is a *spanning subgraph* under $F$.

**Definition A.3** (Graph Connectivity). Let $G = (V, E)$ be a graph. For $v \in V$, let $\mathfrak{N}_0(v) = \{v\}$ and $\mathfrak{N}_1(v) = \mathfrak{N}(v)$. By recursion, define

$$\mathfrak{N}_{i+1}(v) = \mathfrak{N}_i(v) \cup \bigcup_{u \in \mathfrak{N}_i(v)} \mathfrak{N}(u). \tag{A.1}$$

Then $\mathcal{C}(v) = \lim_{i \to \infty} \mathfrak{N}_i(v)$ is called the *connected component* of $v$. If $\mathcal{C}(v) = V$ for any $v \in V$ then $G$ is a *connected graph*.

**Definition A.4** (Reachability). Let $G = (V, E)$ be a graph. We say that two nodes $u, v \in V$ are *reachable* in $G$ if and only if $u \in \mathcal{C}(v)$.

**Definition A.5** (Equivalence Relations and Classes). Let $V$ be a set. Let $\sim \subseteq V \times V$ be a binary relation that is reflexive, transitive, and symmetric. Then $\sim$ is an *equivalence relation on* $V$. Furthermore, for some fixed element $v \in V$, let

$$[v] = \{v' \in V : v \sim v'\}. \tag{A.2}$$

Then $[v]$ is an *equivalence class of* $v$ *under* $\sim$.

**Definition A.6** (Quotient Set). Let $V$ be a set, and let $\sim$ be an equivalance relation on $V$. Then

$$V/\sim = \{[v] : v \in V\} \tag{A.3}$$

is the *quotient set of* $V$ *induced by* $\sim$.

**Definition A.7** (Partition of Sets). Let $V$ be a set. Let $\Pi(V) \subset 2^{2^V}$ such that for all $\pi \in \Pi(V)$

    (i) $\emptyset \notin \pi$,

    (ii) $\pi$ covers $V$, i.e., $\bigcup_{S \in \pi} S = V$,

    (iii) for all $S, S' \in \pi$, if $S \neq S'$, then $S \cap S' = \emptyset$.

Then $\pi$ is a *partition* of $V$, and we call $\Pi(V)$ the *set of all partitions of* $V$.

**Definition A.8** (Refinement of Partitions). Let $V$ be a set, and let $\pi, \pi' \in \Pi(V)$. If for all $S \in \pi$ there exists $S' \in \pi'$ such that $S \subseteq S'$, then we say that $\pi$ is a *refinement* of $\pi'$, denoted by $\pi \sqsubseteq \pi'$. Furthermore, we say that $\pi$ is *finer* than $\pi'$, and equivalently that $\pi'$ is *coarser* than $\pi$.

**Definition A.9** (Hierarchy of Partitions). Let $V$ be a set, and let $\mathcal{H} = \left( \pi_t \in \Pi(V) : \pi_t \sqsubseteq \pi_{t+1} \right)_{t=0}^{T}$. Then $\mathcal{H}$ is a *hierarchy of partitions* ordered by refinement.

**Theorem A.10** (Fundamental Theorem on Equivalence Relations). *Let $V$ be a set, and let $\sim$ be an equivalence relation on $V$. Then $V/\sim \in \Pi(V)$.*

**Definition A.11** (Weighted Graph). Let $G = (V, E)$ be a graph. Let $\chi : E \to \mathbb{R}$ be a function on the edges of $G$. Then $G' = (V, E, \chi)$ is called a *weighted graph*.

## A.2 Hierarchical Graph Partitions

In this section, we construct the main result Definition A.14, which formalizes the construction of hierarchical partitions via monotonic binary edge coloring.

**Proposition A.12** (Partition by Edge Coloring). *Let $G = (V, E, \chi)$ be a weighted graph, and let $\chi \colon E \to \{0, 1\}$ be a binary coloring of the edges such that the edge set*

$$E_\chi = \{\{u, v\} \in E : \chi(u, v) = 1\} \tag{A.4}$$

*is invariant under transitive closure; i.e., $E_\chi^+ = E_\chi$. Then, the coloring $\chi$ induces a partition of $V$ into connected components, where each component is connected via $E_\chi$.*

> **Proof.** From $E_\chi$, we construct a relation $\sim$ on $V$ such that $u \sim v$ if and only if $u$ is reachable by $v$ in the subgraph $G[E_\chi]$. Note that $\sim$ is symmetric since $G$ is undirected, and transitive due to $E_\chi^+ = E_\chi$. Since $v \in \mathcal{C}(v)$ for all $v \in V$, then $v \sim v$ so $\sim$ is necessarily also reflexive. Hence, $\sim$ is an equivalence relation on $V$, and by the fundamental theorem of equivalence relations, the quotient set $V/\sim$ is a partition corresponding to the connected components of $G[E_\chi]$. $\square$

**Proposition A.13** (Hierarchical Partitioning by Monotonic Edge Coloring). *Let $G = (V, E, \chi)$ be a weighted graph, and let $\chi \colon E \times \{0, \dots, T\} \to \{0, 1\}$ be a binary coloring of the edges satisfying*

> *(i) Monotonicity, $\chi(u, v, t) \leq \chi(u, v, t + 1)$ for all $\{u, v\} \in E$ and $t = 0, \dots, T - 1$.*
> *(ii) For each step $t = 0, \dots, T$, the edge set $E_\chi(t) = \{\{u, v\} \in E : \chi(u, v, t) = 1\}$ is invariant under transitive closure, i.e., $E_\chi^+(t) = E_\chi(t)$.*

*Then, $\chi$ induces a hierarchical partition of $V$ ordered by refinement; i.e., the partition induced by $\chi_t$ is a refinement of the partition induced by $\chi_{t'}$ for all $t \leq t'$, where we have that $\chi_t(u, v) = \chi(u, v, t)$ for all $\{u, v\} \in E$.*

> **Proof.** By Proposition A.12, we have that each $\chi_t$ induces an equivalence relation $\sim_t$, partitioning $V$ into equivalence classes at level $t$. Denote this partition by $\pi_t$. We will show that for all $t \leq t'$, the partition $\pi_t$ is a refinement of $\pi_{t'}$. The monotonicity criteria (i) $\chi(u, v, t) \leq \chi(u, v, t + 1)$ implies $E_\chi(t) \subseteq E_\chi(t + 1)$. By induction, $E_\chi(t) \subseteq E_\chi(t')$ for all $t \leq t'$, Since $E_\chi(t) \subseteq E_\chi(t')$, any path using edges in $E_\chi(t)$ is also a path in $E_\chi(t')$. Therefore, if $u \sim_t v$, then $u \sim_{t'} v$. Now, let $[u]_t$ denote the equivalence class of $u$ under $\sim_t$. Then $[u]_t \subseteq [u]_{t'}$ for all $u \in V$ and all $t \leq t'$. Since $\pi_t = V/\sim_t = \{[u]_t : u \in V\}$, we have that $\pi_t \sqsubseteq \pi_{t'}$, as we wanted to show. $\square$

**Definition A.14** (Hierarchical Graph Partition). Let $G = (V, E, \chi)$ be an weighted graph where $\chi : E \times \{0, \dots, T\} \to \{0, 1\}$ is a binary coloring as in Proposition A.13, i.e., monotonic and invariant under transitive closure. Let $\chi(u, v, 0) = 0$ for all $\{u, v\} \in E$ and let $\sim_t$ denote the equivalence relation induced by $E_\chi(t)$ for $t = 0, \dots, T$. Then the sequence

$$G[E_\chi(t)] = (V/\sim_t, E_\chi(t)), \quad t = 0, \dots, T \tag{A.5}$$

is called a *hierarchical graph partition*, where for each $v \in V$ we have that each equivalence class $[v]_t = S \in \pi_t$ denotes a connected region for $\pi_t \in \mathcal{H}$. For notional convenience, we write $G[E_\chi(t)] = G_t = (V_t, E_t)$.

# B Modeling Assumptions and Estimators

In this section, we discuss details regarding methodological assumptions of $\partial$HT from Section 2.4. We cover the i.i.d. Gaussian assumption on the distribution of pixels, show that this approximation has precedence, and empirically verify that this it is a reasonable modeling choice, and derive the estimator for $\mathrm{df}_{\pi_*}$ via atomistic properties of the partition lattice.

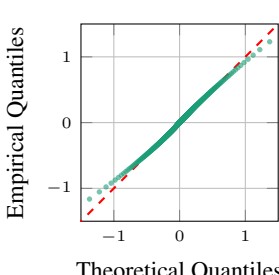 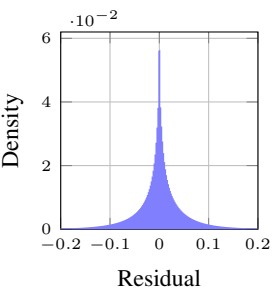

Theoretical Quantiles      Residual

**Table B.1:** Ablation on sensitivity to Gaussian Assumptions.

| Metric | Shape parameter | | |
|--------|-----------------|--------|--------|
| | $b = 2.0$ | $b = 1.0$ | $b = 0.5$ |
| MSE | 0.06 | 0.07 | 0.09 |
| SSIM | 0.60 | 0.57 | 0.55 |

**Figure B.1:** QQ-plot (left) and residual density (right) for pixel residuals in ImageNet-Val for $\partial$HT. The residuals closely follow a Generalized Normal Distribution.

## B.1 Distrubutional Assumptions

A superpixel model can be viewed as a spatially piecewise-constant approximation to image data, constrained by connectivity requirements [68]. Specifically, for an image $x$ a superpixel model $\mathcal{M}_\pi$ is on the form

$$\mathcal{M}_\pi(v) = \mathbf{1}[v \in S]\,\mu_S. \tag{B.1}$$

Given an image defined on pixels $v \in V$, we assume the existence of a partition $\pi_*$ into connected regions that minimize

$$\mathcal{L}_{\text{MSE}} = \frac{1}{|V|} \sum_{v \in V} \|x(v) - \mathcal{M}_{\pi_*}(v)\|^2, \tag{B.2}$$

where $\mathcal{M}_{\pi_*}(v)$ takes constant value $\mu_S$ for all pixels $v \in S$, and the optimal $\mu_S$ is given by the arithmetic mean of pixel intensities within $S$. Given a fixed partition, this choice of estimator is optimal under the Gauss-Markov theorem as the best linear unbiased estimator (BLUE).

Within each region $S$, the estimator $\mu_S$ is BLUE under the assumption that the noise affecting pixels is independent, identically distributed (i.i.d.) with finite variance and zero mean. Further, adopting a Gaussian distribution for pixels conditioned on region membership $x_v \mid S \sim \mathcal{N}(\mu_S, \Sigma_S)$ aligns the squared-error minimization directly with maximum likelihood estimation. Specifically, for diagonal $\Sigma_S$, maximizing the Gaussian log-likelihood corresponds exactly to minimizing within-region variance, connecting clearly with variance-reduction criteria in regression trees and quadtrees [20, 25], supporting our argument in Section 2.4. This is clear from the log-likelihood for a multivariate Gaussian of dimension $d$ for a region $S$, which is given by

$$\log L(\theta_S) = -\frac{|S|}{2}(d\log(2\pi) + \log\det\Sigma_S) - \frac{1}{2}\sum_{v \in S}(x_v - \mu_S)^\mathsf{T}\Sigma_S^{-1}(x_v - \mu_S) \tag{B.3}$$

$$= -\frac{|S|}{2}\Big(d\log(2\pi e) + \log\det\Sigma_S\Big) \tag{B.4}$$

where (B.4) follows by the MLE for $\Sigma_S$ given $\mu_S$.

We emphasize that these assumptions are made explicitly and are not automatically justified by the existence of an optimal partition under variance minimization. The choice of empirical variance as a criterion to evaluate model fit is commonly employed in nonparametric contexts without distributional assumptions, hence the assumption of Gaussianity is not strictly necessary but implicit by the choice of risk minimizer. Claeskens and Hjort [24] show that, even if the candidate models are not parametric distributions, IC approaches remain asymptotically valid in selecting a model that minimizes expected prediction error.

To empirically examine the appropriateness of these assumptions, we evaluated the pixel intensity distributions within representative superpixels over ImageNet1k. Figure B.1 show the residuals follow a generalized Gaussian distribution s.t. $x \mid S \sim \mathcal{GN}(.551, \mu_S, .031)$. As $\mathcal{GN}(2, \mu_S, \Sigma_S) = \mathcal{N}(\mu_S, \Sigma_S)$, the model is reasonable, however the residuals are somewhat sharper and heavier tailed than the approximation.

**Table C.1:** Additional Single Scale Semantic Segmentation mIoU on COCO-Stuff [42] for the original 10k fold. Note the lack of comparative baselines for base capacity models.

| Dataset | Backbone | Method | Size ($\downarrow$) | mIoU ($\uparrow$) |
|---|---|---|---|---|
| COCO10k | Swin-L | SeMask [69] | 640 | 47.4 |
| | ConvNext-L | CAR [70] | 640 | 49.0 |
| | Swin-L | Senformer [71] | 640 | 49.8 |
| | ViT-L | Segmenter [54] | 512 | 47.1 |
| | ViT-B | $\partial$HT + MLP | 512 | 49.2 |

We also assess the robustness to the Gaussian assumption. We trained tokenizers under alternative Generalized Normal distributions with shape parameters $b \in \{2, 1, 0.5\}$, where $b = 2$ corresponds to the Gaussian baseline. Table B.1 shows that the Gaussian assumption yields the best reconstruction quality, despite final residuals being closer to $b \approx 0.551$. This occurs because the distribution is closer to Gaussian in early training iterations, providing initial stability for model fitting.

The Gaussian assumption primarily affects pruning through residual variance estimation; heavier-tailed residuals inflate variance, making the criterion more conservative. Importantly, the complexity penalty remains invariant to distributional misspecification as it depends only on graph topology, and $\partial$HT demonstrates robustness consistent with theoretical results on information criteria under model misspecification [24].

## B.2 Atomistic Properties of $\Pi(G)$

We informally outline some fundamental lattice theory [27] and describe how we can derive an estimate of the degrees of freedom for a partitioned graph under the constraint of connectivity.

The set $\Pi(V)$ is a partially ordered set (poset) under refinement (Definition A.8), and contains two seemingly trivial elements; one is the minimal partition $\bot = \{\{v\} : v \in V\}$, called the *bottom* where all elements of $V$ are individual blocks. Dually, the *top* $\top = \{V\}$, is a partition in which all elements are grouped in a single block. Any finite nonempty subset $V$ will necessarily satisfy $\bot, \top \in \Pi(V)$. In the partition lattice $\Pi(V)$, atoms are defined as the minimal non-trivial partitions that cover the bottom element $\bot$, where each vertex are isolated singletons. Dually, the co-atoms are the elements covered by $\top$ where all vertices comprise a single set. Independent sets of atoms form what is equivalent to a basis (independent sets) in constructing more complex partitions that define each superpixel. Under connectivity in $G$, the atoms of $\Pi(V)$ are precisely the edges $E$ of $G$.

By assumption of a piecewise constant model, the complexity of the model decreases for courser partitions such that the level of complexity is maximal at $\bot$. Then $\mathrm{df}_{\pi_*}$ is necessarily inversely proportional to the number of independent atoms each superpixel encompasses within $\Pi(V)$. Dually, it is necessarily also proportional to the number of possible bipartitions required to form the partition $v' \cup \{v : v \in V, v \notin v'\}$. Unfortunately, this quantity can be considered more or less intractable, however, an estimate can be derived by considering the dualistic nature of the partition lattice.

Recall that $\mathrm{Vol}_G(S)$ for some $S \subseteq V$ is defined as $|\{\{u, v\} \in E : u, v \in S\}|$. Then $\mathrm{Vol}_G(S)$ is the maximal number of atoms required to form $S$, which yields a direct measure of the number of steps between $S$ and $\bot$. This is typically formalized via a rank function $r : \Pi(V) \to \mathbb{Z}_{\geq 0}$ which turns out to be equivalent to the number of atoms in a partition. However, we are instead interested in the number of steps between $S$ and $\top$. Noting that for our construction, we have that $r(\top) = \mathrm{Vol}_G(V) = |E|$ is a maxima, we can estimate degrees of freedom of $S \in \pi_*(V)$ by letting

$$\mathrm{df}_{\pi_*}(S) \approx |E| \cdot \mathrm{Vol}_G(S)^{-1} \tag{B.5}$$

This serves to penalize partitions $S$ that has lower volume and are closer to $\bot$. In effect, the estimate penalizes higher parameter complexity w.r.t. the piecewise constant model of the image by inducing a preference for larger connected regions in the superpixel partition.

## C Extended Results

In the interest of completeness, we include results for COCO-Stuff on the 10k fold to complement Table 3. Note that COCO10k is a preliminary release with much smaller number of datapoints,

**Algorithm D.1** Single Merge Iteration

**Require:** Feature map $f_t \in \mathbb{R}^{N \times C}$
**Require:** Region map $S_t$, Edge set $E_t$
**Require:** Similarity kernel $\kappa(\cdot, \cdot)$
**Require:** Information criteria $\mathrm{IC}(\cdot, \cdot)$
**Ensure:** Updated feature map $f_{t+1}$
**Ensure:** Updated region map $S_{t+1}$
**Ensure:** Updated edge set $E_{t+1}$
**Ensure:** IC penalty $\mathcal{L}_{\mathrm{IC}}$
  $E_{\max} \leftarrow \emptyset$
  **for** $v \in S_t$ **do**
    $E_{\max}[v] \leftarrow \arg\max_{u:\{u,v\} \in E_t} \kappa(u, v)$
  **end for**
  $S_{t+1} \leftarrow \textsc{ConnectedComponents}(E_{\max})$
  $f_{t+1} \leftarrow \textsc{zeros}(N' \times C)$
  **for** $u \in S_{t+1}$ **do**
    **for** $v \in \{v \mid S_{t+1}(v) = u\}$ **do**
      $v_{\max} \leftarrow E_{\max}[v]$
      $w \leftarrow |S_t(v)| / |S_{t+1}(u)| \cdot \kappa(v, v_{\max})$
      $f_{t+1}[u] \leftarrow f_{t+1}[u] + w \cdot f_t[v]$
    **end for**
  **end for**
  $E_{t+1} \leftarrow \textsc{UpdateEdges}(S_{t+1})$
  **for** $u \in S_{t+1}$ **do**
    $\mathcal{L}_{\mathrm{IC}}[u] \leftarrow \mathrm{IC}(f_{t+1}[u], E_{t+1})$
  **end for**
  **return** $f_{t+1}, S_{t+1}, E_{t+1}, \mathcal{L}_{\mathrm{IC}}$

**Algorithm D.2** Feature Extraction

**Require:** Image tensor $x \in \mathbb{R}^{B \times C \times H \times W}$
**Require:** Region features $f \in \mathbb{R}^{N \times C'}$
**Require:** Region map $S$
**Require:** Grid resolution $q \in \mathbb{N}$
**Require:** Positional resolution $p \in \mathbb{N}$
**Require:** Projection matrix $W \in \mathbb{R}^{C \times C'}$
**Require:** Background token $\beta \in \mathbb{R}^{C \times q \times q}$
**Require:** Mixing weight $\lambda \in [0, 1]$
**Ensure:** Tokenized features $F \in \mathbb{R}^{N \times C \times q \times q}$
**Ensure:** Kernel pos. features $P \in [0, 1]^{N \times p \times p}$
  **for** $u \in S$ **do**
    $\mu_u \leftarrow \mathrm{mean}_{v \in u}(x[v])$
    $\hat{\mu}_u \leftarrow W \cdot f[u]$
    **for** each pixel $p \in u$ **do**
      $\hat{x}[p] \leftarrow x[p] + \hat{\mu}_u - \mu_u$
    **end for**
  **end for**
  $F \leftarrow \textsc{zeros}(N \times C \times q \times q)$
  $P \leftarrow \textsc{zeros}(N \times p \times p)$
  **for** $u \in S$ **do**
    $M \leftarrow \textsc{DownsampleMask}(S = u, q \times q)$
    **for** $(i, j) \in q \times q$ **do**
      $s \leftarrow \textsc{BilinearSample}(\hat{x}, \mathrm{bbox}(u), i, j)$
      $m \leftarrow M[i, j]$
      $s_{\mathrm{mix}} \leftarrow \lambda \cdot s + (1 - \lambda) \cdot \beta[:, i, j]$
      $F[u, :, i, j] \leftarrow m \cdot s + (1 - m) \cdot s_{\mathrm{mix}}$
    **end for**
    $P[u] \leftarrow \textsc{KernelPosEmbed}(S = u, p \times p)$
  **end for**
  **return** $F, P$

**Figure D.1:** Core algorithms for $\partial$HT. Left: Single iteration of hierarchical vertex merging with kernel-weighted aggregation. Right: Differentiable feature extraction with mean-injection and adaptive masking.

which were updated to include the full COCO164k fold at a later time. Consequently, there are fewer baselines available. To the best of our knowledge, out of the works reporting results on COCO10k, there are no instances of base- or small capacity models available. Nevertheless, we include relevant results for COCO10k in Table C.1, which illustrate that $\partial$HT performs relatively well compared to larger models with larger capacity (305M parameters for large (L) compared to 87M for base (B) capacity models).

# D  Implementation and Training Details

We provide a full overview of our experimental setup and training configuration. Training and inference was performed on AMD MI250x and Nvidia A100. We detail central algorithms in Fig. D.1, and provide code and checkpoints in our GitHub repo. Our experiments were conducted as follows:

- **Tokenizer Pretraining**: $\partial$HT modules were pretrained to optimally reconstruct images from ImageNet1k, using (7). We train for 10 epochs using AdamW with 1e-4 learning rate and 1e-2 weight decay, but find that performance saturates between epoch 5–6. We test the performance with different hyperparameter settings—cf. Table 6.

- **Retrofitting**: We select three baseline models trained exclusively on ImageNet1k. Each model is then retrofitted with our pretrained tokenizer, using the configuration in Table D.1(b). Models are fine tuned with layer-wise learning rate decay of 0.65 [72], which improves learning for earlier layers. We evaluate the models over various downstream tasks, yielding the results in Table 1.

- **Scale Invariance**: Given how the baseline ViT-B32 model produces very few regions, we perform a comparative evaluation by adding more fine-grained control over the number of tokens over different image resolutions. We add merging mechanisms which serves to limit the total number of tokens in a model, and evaluate baselines and retrofitted models over various image sizes. The results are provided in Figure 6.

**Table D.1:** Configuration parameters for different stages

**(a) Pretraining**

| config | value |
|---|---|
| batch size | 2048 |
| epochs | 400 |
| dataset | ImageNet1k |
| img.size | $192 \times 192$ |
| pos.emb. | $16 \times 16$ |
| loss fn. | CE (0.1 smooth.) |
| optimizer | LAMB |
| lr.sched. | cos.decay (5 w.u.) |
| lr (start / base / stop) | $3e-3$ / $3e-7$ / $1e-6$ |
| momentum | 0.9 |
| dropout path | 0.1 (S) / 0.2 (B) |
| opt. $\epsilon$ | $1e-7$ |
| cutmix $\alpha$ | 1.0 |
| augment | rand.aug. / aug3 |

**(b) Tokenizer Retrofitting**

| config | value |
|---|---|
| batch size | 2048 |
| epochs | 100 |
| dataset | ImageNet1k |
| img.size | $192 \times 192$ |
| pos.emb. | $16 \times 16$ |
| loss fn. | CE (0.1 smooth.) |
| optimizer | LAMB |
| lr.sched. | cos.decay (5 w.u.) |
| lr (start / base / stop) | $1e-7$ / $6e-5$ / $1e-6$ |
| momentum | 0.9 |
| dropout path | 0.1 (S) / 0.2 (B) |
| opt. $\epsilon$ | $1e-8$ |
| augment | rand.aug. / aug3 |
| llrd | 0.65 |

**(c) Finetuning**

| config | value |
|---|---|
| batch size | 512 |
| epochs | 100 |
| dataset | ImageNet1k |
| img.size | $224 \times 224$ |
| pos.emb. | $24 \times 24$ |
| loss fn. | CE (0.1 smooth.) |
| optimizer | AdamW |
| lr.sched. | cos.decay (5 w.u.) |
| lr (start / base / stop) | $1e-6$ / $1e-5$ / $1e-5$ |
| dropout path | 0.1 (S) / 0.2 (B) |
| opt. $\epsilon$ | $1e-8$ |
| augment | rand.aug. / aug3 |
| llrd | 0.9 |

**(d) Segmentation Finetuning**

| config | value |
|---|---|
| batch size | 512 |
| epochs | 400 |
| dataset | COCO-Stuff, ADE20k |
| img.size | $512 \times 512$ |
| pos.emb. | $48 \times 48$ |
| loss fn. | BCE + Focal |
| optimizer | AdamW |
| lr.sched. | cos.decay (5 w.u.) |
| lr (start / base / stop) | $1e-6$ / $1e-5$ / $1e-5$ |
| dropout path | 0.1 (S) / 0.2 (B) |
| opt. $\epsilon$ | $1e-8$ |
| augment | rand.aug. / aug3 |
| crop scale / ratio | (0.5, 1.0) / (0.8, 1.2) |
| llrd | 0.85 |

- **Full Training**: We extend these experiments by evaluating a full training procedure, following the training process outlined by Touvron et al. [37], Steiner et al. [38], notably without the use of MixUp [73] augmentation, as blended images produces inaccurate boundaries for learning coherent regions. Following previous works [15], models trained from scratch apply gradient histogram features. As is generally recommended, the training was performed in two steps, outlined in Table D.1(a) and Table D.1(c) respectively. The results are featured in the lower half of Table 1.

- **Segmentation Fine Tuning**: Given our fully trained $\partial$HT models, we perform fine tuning for semantic segmentation. We replace each head with a single hidden-layer MLP with a hidden ratio of $4\times$. The fine tuning is performed using the configuration in Table D.1(d), and results are reported in Table 3.

- **Zero-shot Salient Segmentation**: We evaluate our fine-tuned classification model on zero-shot salient segmentation. We emphasize that the model has not been trained for this task. Following Wang et al. [58], we compute the graph Laplacian of the token representations, and compute a bipartition using the Fiedler vector. Foreground masks are selected by passing masked tokens through the transformer, and selecting the mask that sees the least drop in performance under occlusion. Results are featured in Table 4.

- **Image Vectorization**: Our $\partial$HT tokenizer provide high fidelity superpixels, which can be directly applied for image vectorization out-of-the-box. From our pretrained tokenizer we extract both an optimal, as well as a low granularity partition, noting that lower granularity partitions has much fewer superpixels. We then extract paths for each superpixel in the lower granularity region, and layer high granularity paths on top using potrace [61], resulting in an SVG image. We compare results with quantitative baselines in Table 5.

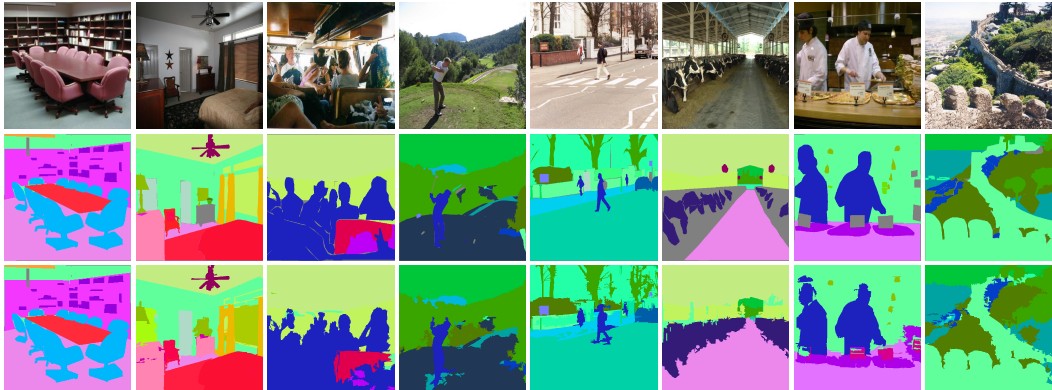

**Figure E.1:** Segmentation examples for ∂HT over ADE20k, showing fine grained segmentation labels only using a simple MLP head without upscaling. *Top*: Original images (512 × 512). *Middle*: Annotated target images. *Bottom*: Predicted labels from ∂HT.

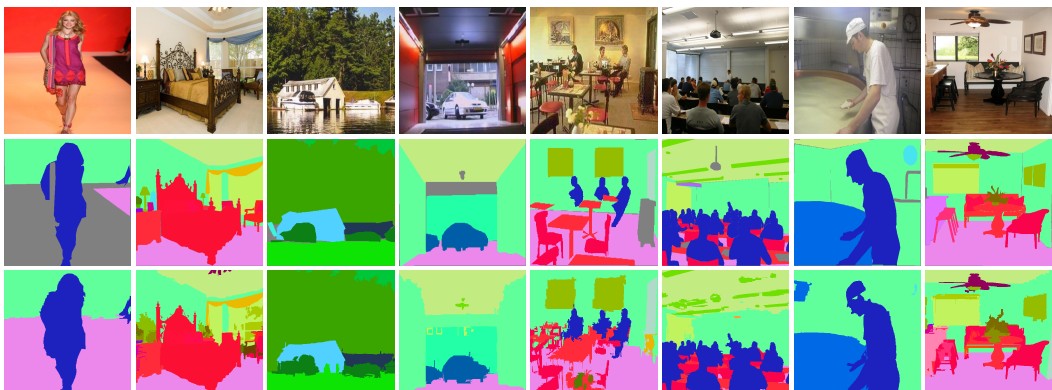

**Figure E.2:** More examples of semantic segmentation on ADE20k. *Top*: Original images. *Middle*: Annotated target images. *Bottom*: Predicted labels from ∂HT.

## E   Qualitative Results and Visualizations

In this section, we extend the visualizations and results from dense predictions and image vectorization. Figures E.1 and E.2 shows a selection of segmentation results on ADE20k validation. ∂HT performs generally well, particularly considering that the predictions are produced with single scale, i.e. we only use the last level of tokens to produce predictions. The results are in large part due to the granularity of the model, and the ability to adapt the partitions to the image data. The example in the second column of Figure E.2 shows an example where the granularity of the prediction is arguably better than annotations. The examples also demonstrate some typical failure cases, particularly with undersegmentation for people in the distance, e.g. column four and five of Figure E.1.

In Figure E.3, we show more raster-to-vector graphics conversions on example images from COCO-Val. We emphasize that our method produces high quality results, despite a comparatively simpler approach to image vectorization. Unlike other approaches [59, 62], our method does not yield differentiable paths, and does not optimize the vector graphics for each individual image. Instead, we simply use the results of our tokenizer, trained with unrelated downstream tasks, to produce vectorized images.

Figure E.5 illustrates the properties of superpixel hierarchies constructed with different images, showing how over- and under-segmentation necessitates adaptive model selection mechanisms, similar to mixed-scale tokenization strategies. By selecting partitions dynamically, ∂HT can adapt to image data over different scales, and adapt regions to fit image content with pixel level granularity.

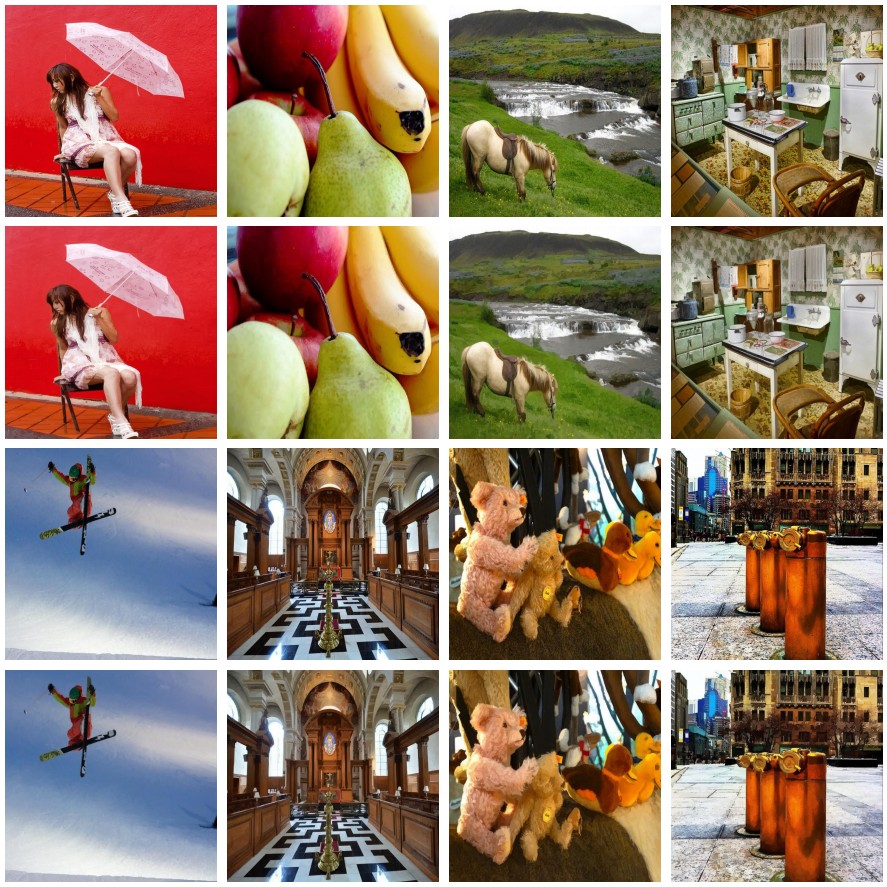

**Figure E.3:** Examples of vectorized images using $\partial$HT. The top rows shows original images, while the bottom rows shows the results of image vectorization. The vectorized images contain (on average) $\sim 5000$ paths.

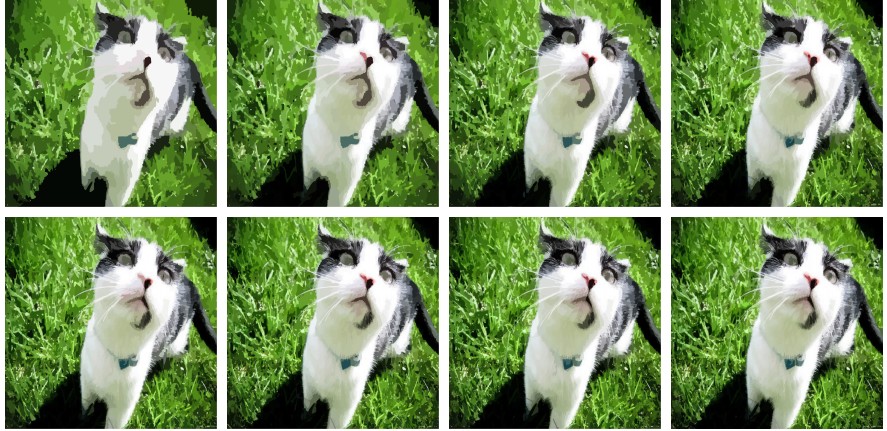

**Figure E.4:** Vectorization results with different numbers of paths, increasing from $\sim 500$ (left) to $\sim 5000$ (right).

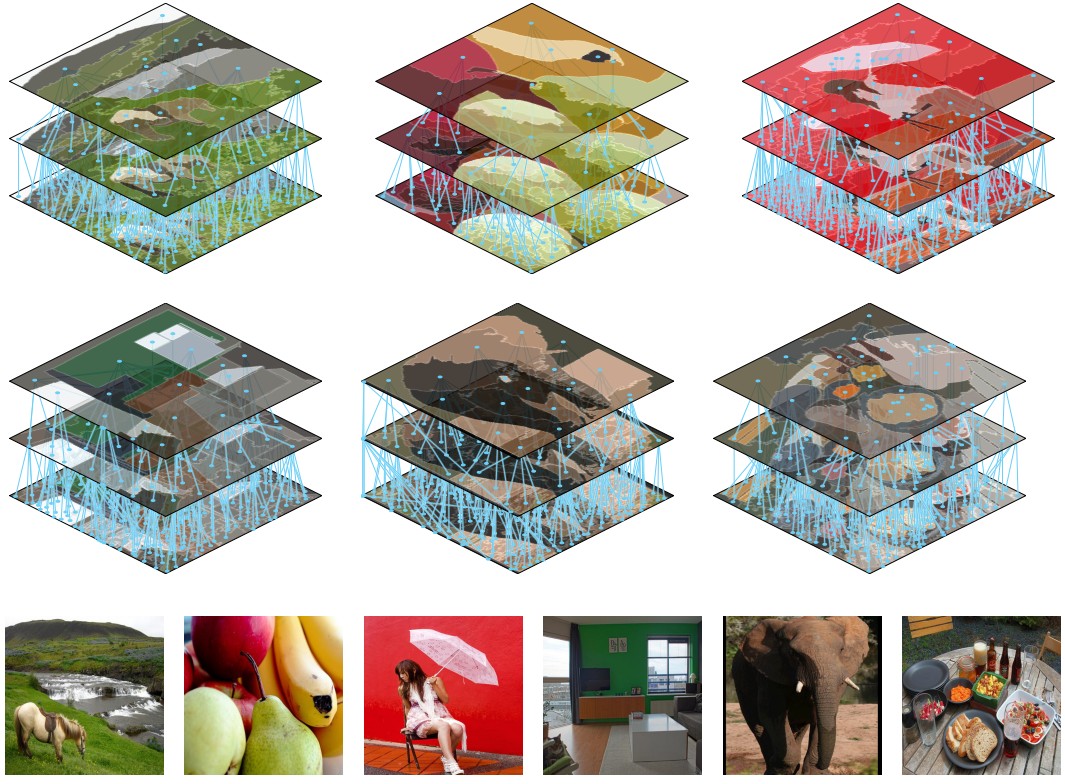

**Figure E.5:** *Top:* Detailed view of superpixel hierarchies for different images, cf.—Fig. 2. Note that the higher levels often yield oversegmented regions, while lower levels yield undersegmented regions. By using model selection with information criteria, our model selects the most informative tokens over the full hierarchy. *Bottom:* Original images, included for reference.

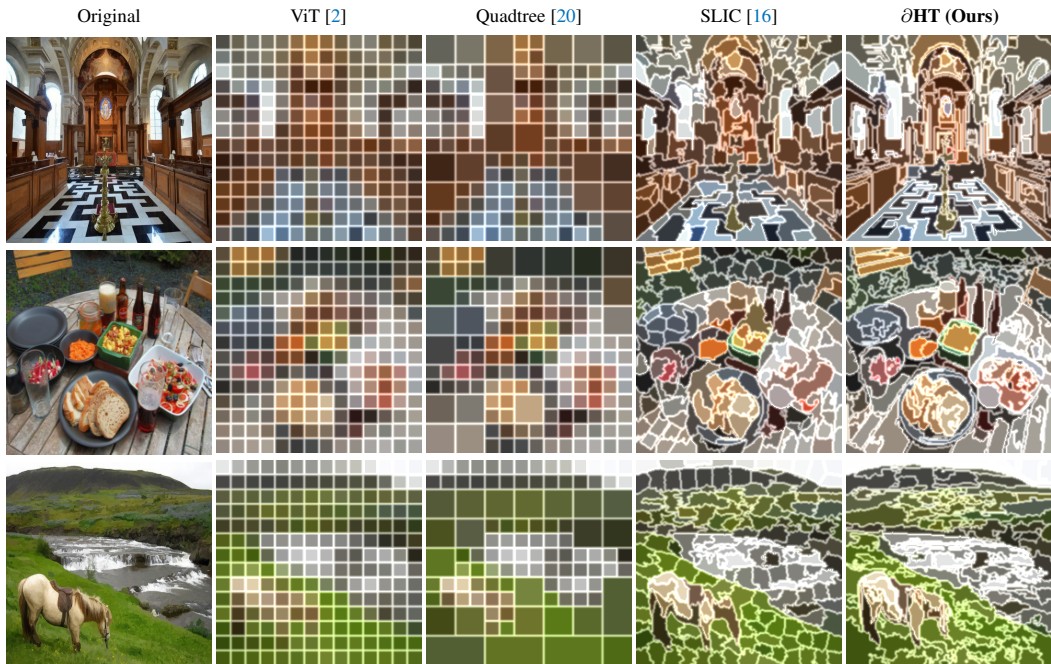

**Figure E.6:** Comparison of spatial granularity in tokenization methods. Our proposed $\partial$HT (right) provides an end-to-end learnable framework for tokenization.

