# OpenReview forum: "Differentiable Hierarchical Visual Tokenization"
_NeurIPS.cc/2025/Conference — NeurIPS 2025 spotlight_

### Official Review · Reviewer_foiG · 2025-06-24

**Clarity:** 2
**Significance:** 3
**Originality:** 3
**Rating:** 5
**Confidence:** 3

**Summary:**

Vision Transformers (ViTs) traditionally use fixed grid-based image patches for tokenization, which ignores semantic object boundaries and spatial hierarchies. This paper introduces ∂HT (Differentiable Hierarchical Tokenization), an end-to-end learnable tokenizer that adapts to image content at pixel-level granularity while maintaining backward compatibility with existing ViTs.

**Questions:**

It is also essential to compare the model's parameter count and computational time in the experiments.

The "decoder-free" segmentation uses a simple MLP head. Is ∂HT’s performance bottleneck the tokenizer or the MLP head?

The IC formulation assumes i.i.d. Gaussian pixels, but Fig. B.1 shows heavier-tailed residuals. How sensitive is ∂HT to this assumption?

**Ethical Concerns:**

["NO or VERY MINOR ethics concerns only"]

**Final Justification:**

The authors' rebuttal has fully resolved all my concerns. I have therefore raised my overall rating of the paper.

**Limitations:**

yes

**Quality:**

3

**Strengths And Weaknesses:**

Strengths:

a). The idea makes sense. ∂HT bridges the semantic gap in visual tokenization by unifying adaptability, differentiability, and modularity. It demonstrates that tokenization can be a learnable component in ViTs, enabling dynamic, content-aware representations without architectural overhaul.

b). Rigorous Methodology. The paper introduces a novel differentiable tokenization framework (∂HT) with well-defined theoretical foundations. The hierarchical partitioning via monotonic edge coloring is mathematically sound.

c). Comprehensive Experiments. Extensive evaluation across tasks  and datasets demonstrates robustness. Ablation studies validate design choices (e.g., kernel selection, positional embedding resolution).


Weaknesses:

a). The writing is not very clear. The organization of the sections is a bit messy, such as 1.1 and 4.1.

b). It is also essential to compare the model's parameter count and computational time in the experiments.

c). In the Main Results of Table 1, improvements over baselines are modest,  if not negative, raising questions about practical impact.

---

> ### Author Rebuttal · Authors · 2025-07-30
>
> ## Response to foiG
>
> We appreciate the reviewers time in reviewing our submission. Their constructive comments have contributed towards refining and enhancing our manuscript.
>
> ---
>
> ### Strengths
>
> > *The idea makes sense. ∂HT bridges the semantic gap in visual tokenization by unifying adaptability, differentiability, and modularity (...) demonstrates that tokenization can be a learnable component in ViTs*
>
> > *well-defined theoretical foundations (...) hierarchical partitioning via monotonic edge coloring is mathematically sound.*
>
> > *Extensive evaluation across tasks and datasets demonstrates robustness. Ablation studies validate design choices*
>
> We thank the reviewer for their summary, and appreciate their comments on the motivation, theoretical grounding, and empirical evaluation of our approach.
>
> ---
>
> ### Weaknesses
>
> > *the organization of the sections is a bit messy, such as 1.1 and 4.1.*
>
> We understand why the reviewer found these two sections a little unclear, since each combines different purposes (motivation and related work in 1.1, further work and limitations in 4.1) into single sections. We will separate these into different sections in our final revision.
>
> ---
>
> > *essential to compare the model's parameter count and computational time in the experiments*
>
> We agree on both points; parameter counts and computational overhead are indeed important metrics that would be of interest to readers.
>
> In terms of parameter count, dHT adds 53k parameters compared to the DEiT3-B16 baseline, which is 4 orders of magnitude less than the total number of parameters (.06%). We will include explicit parameter counts for all baselines and models, and report the results in our final revision.
>
> Given the adaptive nature of dHT, token counts differs depending on the image. Following reviewer rgEH's suggestion, we report *tokens-per-second* for dHT compared to baseline ViTs with base capacity, along with average tokens per image. This provides balanced evaluation of the computational overhead associated with dHT. We will include the results in the camera ready version of our manuscript.
>
> | Capacity| Resolution| Tokenizer | Tok./s. | Tok./im.|
> |---------|-----------|-----------|---------|---------|
> | S16     | 224       | Patch     | 192.3k  | 197.0   |
> | S16     | 224       | dHT       |  94.2k  | 240.3   |
> | B16     | 224       | Patch     |  89.7k  | 197.0   |
> | B16     | 224       | dHT       |  68.2k  | 245.7   |
> | S16     | 384       | Patch     | 112.4k  | 577.0   |
> | S16     | 384       | dHT       |  58.1k  | 494.2   |
> | B16     | 384       | Patch     |  61.6k  | 577.0   |
> | B16     | 384       | dHT       |  48.9k  | 431.8   |
>
> *Throughput is computed over ImageNet-val (50k images), averaged over 4$\times$MI250x* with `fp32` precision.
>
> The results show that dHT incurs a moderate reduction in tokens processed per second compared to the standard patch tokenizer, as we discuss in our limitations section [L303]. However, as mentioned in our paper, the reduce in throughput is significantly lower at higher capacities and resolutions, which shows promise for high resolution tasks and video.
>
> ---
>
>
> > *In (...) Table 1, improvements over baselines are modest, if not negative, raising questions about practical impact*
>
> We acknowledge that improvements vary by task and model configuration, and therefore opted to characterize our results as *competitive* while clearly specifying that dHT does not necessarily improve performance in *every* comparison [L199-L206].
>
> To put the results in context, Table 1 summarizes 30 comparisons on top-1 linear accuracy and kNN performance across 6 datasets and five model configurations. In the majority of cases, dHT is competitive or better than the baselines. Specifically, for top-1 accuracy, *dHT outperforms the patch baseline in 80% (24/30) cases*, and *70% (42/60) when kNN is included*. The mean improvement is +3.4 pp. for top-1 and +2.9 pp. when including kNN.
>
> When dense-prediction tasks are included (Tables 3–5), we note a trend of consistent performance gains on instance level as well as dense downstream tasks.
>
> ---
>
> ### Questions
>
> > *The "decoder-free" segmentation uses a simple MLP head. Is ∂HT’s performance bottleneck the tokenizer or the MLP head?*
>
> This is a very interesting question. We believe there are gains to be had in both MLP head as well as the tokenizer. In our experiments, we opted to keep the architecture consistent between the tasks (classification and segmentation), and the capacity of the CNN in the tokenizer could potentially be expanded to improve performance in dense tasks. Likewise, we do comparision on single scale (output) baselines for the MLP, and results could potentially be improved by expanding to a multi-scale approach (i.e., concatenating outputs from multiple blocks) which would expand the MLP capacity.
>
> We emphasize that the intent of the decoder-free segmentation head is to measure the representational quality of tokens "out-of-the-box". More expressive decoders are likely to improve both baselines and dHT proportionally. If we were to select between the two, we believe adding more capacity to the MLP head is more likely to improve segmentation results.
>
> ---
>
> > *The IC formulation assumes i.i.d. Gaussian pixels, but Fig. B.1 shows heavier-tailed residuals. How sensitive is ∂HT to (Gaussian) assumption?*
>
> In Appendix B.1, we connect Gaussian assumptions to empirical variance as a risk minimizer, and discuss the validity of non-parametric (e.g., ERM) models for information criteria (IC). Claeskens and Hjorth [24] also discuss robustness of IC under misspecification, showing that information criteria are *relatively robust to misspecification*.
>
> In our case, the Gaussian assumption is equivalent to a Generalized Normal ($\mathcal{GN}$) but with a higher shape parameter $b$ than the final residuals show. The choice of Gaussian likelihood only affects pruning through the residual variance estimate. While heavier‑tailed residuals inflate variance the effect is that the pruning is more moderate, making the IC more selective. The complexity penalty itself is immune to misspecification, but adding a penalty term for local kurtosis could potentially correct for this at the cost of computing kurtosis per region.
>
> To estimate dHT's robustness to the Gaussian assumption, we train tokenizers with $b \in \{2,1,.5\}$ following the ablations in Table 6. The results yields slightly worse results compared to the Gaussian baseline ($b=2$).
>
> | Metric | $b=2$    | $b=1$     | $b=.5$    |
> |--------|----------|-----------|-----------|
> | MSE    | 0.06     | 0.07      | 0.09      |
> | SSIM   | 0.60     | 0.57      | 0.55      |
>
> Given that the distribution of residuals end up closer to $b=0.551$, this may initially seem somewhat surprising. However, we find that the distribution is closer to standard Gaussian ($b = 2$) in *early iterations of training*, providing initial stability in fitting the model.
>
> To directly answer the reviewer's question; *dHT is not particularly sensitive to assumptions of Gaussianity*. We thank the reviewer for an engaging question, and will include this discussion in our final revision.

---

> > ### Comment · Reviewer_foiG · 2025-08-05
> >
> > Thank the authors for their response. The authors' response has fully resolved all my concerns.

---

> ### Author Response · Authors · 2025-08-07
>
> Thank you for your feedback. We appreciate you letting us know that our response resolved your concerns.

---

### Official Review · Reviewer_hZMf · 2025-06-30

**Clarity:** 2
**Significance:** 3
**Originality:** 3
**Rating:** 5
**Confidence:** 2

**Summary:**

This paper proposes a novel adaptive differentiable hierarchical tokenizer for ViT models, named $\partial$HT. The method first extracts pixel features using a lightweight CNN module, constructs a hierarchical division of pixels via a graph-based paradigm, searches for the best partition using information criteria, and finally aggregates features through a specialized extraction technique. The tokenizer supports various scenarios, including retrofitting pre-trained patch-based ViTs and training from scratch. Extensive experiments demonstrate the effectiveness of the proposed method.

**Questions:**

1. I am curious about the function $f$ in Eq. 2 and Eq. 5. What is the relationship between $\bar{f}_1$ ​ and $f_1$? Is Eq. 2 just a special case of Eq. 5?
2. I am also wondering whether the performance could be further improved if the averaging operation in Eq. 2 and Eq. 5 were replaced with a learnable weighted aggregation, similar to that used in graph neural networks.

**Ethical Concerns:**

["NO or VERY MINOR ethics concerns only"]

**Final Justification:**

I believe the authors' response has resolved my concerns and questions well. And I hope the authors can further refine their work in the camera ready version as they promised.

**Limitations:**

Yes.

**Paper Formatting Concerns:**

No.

**Quality:**

3

**Strengths And Weaknesses:**

### Strength:
1. This paper is highly complete and rigorous. The limitations and future directions are discussed in detail as well. It further explores the feasibility of end-to-end differentiable approaches based on existing irregular tokenizers, reducing the reliance on heuristic rules and adopting a more flexible data-driven design. Moreover, the limitations section discusses various aspects from multiple perspectives, making it quite comprehensive.

2. Extensive experiments and ablations demonstrate the effectiveness of the proposed method. The experiments compare the proposed method with the canonical tokenizer across tasks of different granularities, and the visualization results are impressive.


### Weakness:
The organization of this paper can be further improved for better understandability. Specifically:
  - the "High Level Overview" in Section 2 is not comprehensive enough; instead of only listing what each step does, it should present a clearer logical flow, potentially moving some contents from the following subsections upward, as the overall logic is currently only reflected in Figure 2 and should also be articulated in the text.
  - A clearer explanation of the computation logic and algorithm implementation in Section 2.3 should be provided.
  - Additionally, while conceptual, symbolic, and formulaic definitions are necessary, it would be better to minimize the introduction of new definitions where possible.

---

> ### Author Rebuttal · Authors · 2025-07-30
>
> ## Response to hZMf
>
> Thank you for your time invested in reviewing our work. We appreciate your comments, and look to incorporate your suggestions to improve the quality of our manuscript.
>
> ---
>
> ### Strengths
>
> > *This paper is highly complete and rigorous (...) limitations and future directions are discussed in detail (...) discusses various aspects from multiple perspectives, making it quite comprehensive*
>
> > *Extensive experiments and ablations demonstrate the effectiveness of the proposed method (...) compare the proposed method with the canonical tokenizer across tasks of different granularities, and the visualization results are impressive*
>
> We appreciate the reviewer’s positive remarks and their summary of the key strengths of our work.
>
> ---
>
> ### Weaknesses
>
> > *the "High Level Overview" in Section 2 is not comprehensive enough (...) instead of only listing what each step does, it should present a clearer logical flow*
>
> We appreciate the reviewer highlighting the potential gap in our presentation, and agree that the overview should be improved.
>
> In the camera ready version, we will restructure subsection 2.1 to include a more conceptual overview, briefly recalling the problem statement from the introduction before introducing the high level concept of the main idea.
>
> This will bring the logical flow implicit in Figure 2 into the prose. Steps (ii)–(iv) will be previewed in one paragraph each, with forward references to their detailed subsections.
>
> ---
>
> > *a clearer explanation of the computation logic and algorithm implementation in Section 2.3 should be provided*
>
> We agree that expanding on core logic and algorithmic steps could improve the clarity of our work, and make it more available to readers.
>
> In addition to the code in the repo, we add a boxed “Algorithm 1” pseudocode showing a single iteration of the merge loop.
>
> ```
> ────────────────────────────  Algorithm 1  —  Single merge iteration
> Inputs :    feature map          f_t ∈ ℝ^{N×C}
>             region map           S_t
>             edge set             E_t
> Parameters: similarity kernel    κ(·,·)
>             information criteria IC(·,·)
> Outputs:    updated feature map  f_{t+1}
>             updated region map   S_{t+1}
>             updated edge set     E_{t+1}
>             IC penalty           ℒ_IC
>
>  1: E_max ← ∅                                           # init empty merge graph
>  2: for v in S_t:
>  3:     E_max[v] ← argmax_{u: {u,v} ∈ E_t} κ(u, v)
>
>  4: S_{t+1} ← ConnectedComponents(E_max)                # N → N' merged regions
>  5: f_{t+1} ← zeros(N'×C)                               # init new feature map
>
>  6: for u in S_{t+1}:                                   # for each new region
>  7:     for v in {v | S_{t+1}(v) == u}:                 # for each child region
>  8:         v_max ← E_max[v]
>  9:         w ← |S_t(v)| / |S_{t+1}(u)| · κ(v, v_max)
> 10:         f_{t+1}[u] += w · f_t[v]                    # weighted aggregation
>
> 11: E_{t+1} ← UpdateEdges(S_{t+1})
>
> 12: for u in S_{t+1}:
> 13:     ℒ_IC[u] ← IC(f_{t+1}[u], E_{t+1})
>
> 14: return f_{t+1}, S_{t+1}, E_{t+1}, ℒ_IC
> ────────────────────────────────────────────────────────────────────────────────
> ```
>
> Additionally, we add a boxed "Algorithm 2" pseudocode to clarify details on feature extraction.
>
> ```
> ───────────────────────  Algorithm 2  —  Feature extraction
> Inputs :    image tensor            x ∈ ℝ^{B×C×H×W}
>             region features         f ∈ ℝ^{N×C′}
>             region map              S                    # pixel → region index
> Parameters: grid resolution         q ∈ ℕ
>             positional resolution   p ∈ ℕ
>             projection matrix       W ∈ ℝ^{C×C'}         # learnable
>             background token        β ∈ ℝ^{C×q×q}        # learnable
>             mixing weight           λ ∈ [0,1]            # learnable
> Outputs :   tokenized features      F ∈ ℝ^{N×C×q×q}
>             kernel pos. features    P ∈ [0,1]^{N×p×p}
>
>  1: for u in S:
>  2:     μ_u ← mean_{v ∈ u}(x[v])                         # empirical mean
>  3:     μ̂_u ← W ⋅ f[u]                                   # predicted mean
>  4:     for each pixel p in u:
>  5:         x̂[p] ← x[p] + μ̂_u - μ_u                      # Equation (6)
>
>  6: F ← zeros(N×C×q×q)
>  7: P ← zeros(N×p×p)
>
>  8: for u in S:
>  9:     M ← DownsampleMask(S == u, q×q)                  # foreground occupancy
> 10:     for (i,j) in q×q:
> 11:         s ← BilinearSample(x̂ , bbox(u), i, j)
> 12:         m ← M[i,j]
> 13:         s_mix ← λ · s + (1−λ) · β[:,i,j]             # pixel mask blending
> 14:         F[u,:,i,j] ← m · s + (1−m) · s_mix           # Equation (8)
> 15:     P[u] ← KernelPosEmbed(S == u, p×p)
>
> 16: return F, P
> ────────────────────────────────────────────────────────────────────────────────
> ```
>
> We thank the reviewer for their suggestion, and agree that additional pseudocode helps communicate the computational logic in our manuscript. It may also serve as a helpful companion to the code in the project repo.
>
> ---
>
> > *while conceptual, symbolic, and formulaic definitions are necessary, it would be better to minimize the introduction of new definitions*
>
> We will audit Section 2 and the appendix and remove one-off symbols or terms that appear only once or twice. When a quantity is used in just a single derivation, we will look to reuse existing symbols. Any lattice-specific terms that are needed only for the proofs (join, meet, rank) will stay in Appendix A.
>
> We thank the reviewer for their comments, and their contribution to improving the clarity of our work.
>
> ---
>
> ### Questions
>
> > *I am curious about the function in Eq. 2 and Eq. 5. What is the relationship between $\bar f_1$ and $f_1$ Is Eq. 2 just a special case of Eq. 5?*
>
> Yes, Eq. 2 is the *uniform-kernel* instance of the more general update in Eq. 5.
>
> $\bar f_{1}(v)$ (Eq. 2) is the pure arithmetic mean of the raw pixel embeddings $f_{0}(u)$ over the first connected component $S=C_{1}(v)$, which demonstrates how aggregation is performed by Aasan et al. [15]. $f_{1}(v)$ in Eq. 5 shows our proposed approach with dHT.
>
> At $t=0$, every “region” $S_{0}$ is a single pixel, so $|S_{0}|=1$, the ratio $|S_{0}|/|S_{1}|=1/|S_{1}|$. For later levels ($t\ge 1$) Eq. 5 keeps accumulating information with *kernel-weighted averages* and the size ratio $|S_{t}|/|S_{t+1}|$ so that features stay unbiased while still carrying the similarity signal. The extra weighting is what the barless $f_{t+1}$ symbol is meant to capture.
>
> We see how the this notation could potentially be unclear to readers, and we will more clearly specify that the vertex merging in Eq. 2 is a general case of Eq. 5.
>
> ---
>
> > *I am also wondering whether the performance could be further improved if the averaging operation in Eq. 2 and Eq. 5 were replaced with a learnable weighted aggregation, similar to that used in graph neural networks.*
>
>
> That is an interesting suggestion. Eq. 5 already performs an *edge-weighted aggregation*
>
> $$
> f_{t+1}(v)=\sum_{u\in S_t(v)}
> \tfrac{|S_t(u)|}{|S_{t+1}(v)|}\,
> \kappa(u,u_{\max})\,
> f_t(u),
> $$
>
> where the *kernel weight* $\kappa(u,u_{\max})$ is recomputed from the current feature vectors at every training step (Section 3.1). Because those features receive gradients, the weights are *implicitly learnable*. Introducing an *explicitly parameterised attention kernel* would be an interesting extension, but a few practical constraints arise.
>
> - *Symmetry:* Our setup assumes $\kappa(u,v)=\kappa(v,u)$. A key–query attention map would break that symmetry, doubling the number of kernel evaluations and requiring an additional normalisation step to keep the update stable.
> - *Positive-definiteness:* Eq. 1 assumes $\kappa$ is a positive semi-definite (PSD) similarity kernel so that the edge set $E_{\max}$ remains well-defined. A freely parametrized kernel should preserve PSD to keep the theoretical guarantees (Appendix A) of the hierarchy construction.
>
> We agree that exploring parametric extensions (e.g. an adaptive PSD kernel or a learned Mahalanobis metric) is an appealing avenue for future work, and will add a discussion on learnable extensions for weighted aggregation in the camera ready version.

---

> > ### Comment · Reviewer_hZMf · 2025-08-05
> > **Reply to authors' response**
> >
> > I believe the authors' response has resolved my concerns and questions well. And I hope the authors can further refine their work in the camera ready version as they promised.

---

> > > ### Author Response · Authors · 2025-08-07
> > >
> > > Many thanks for acknowledging that our response resolved your concerns. We will integrate the additional clarifications and enhancements into the final version as promised.

---

### Official Review · Reviewer_rgEH · 2025-07-02

**Clarity:** 3
**Significance:** 4
**Originality:** 4
**Rating:** 5
**Confidence:** 4

**Summary:**

The authors propose a hierarchical approach to image tokenization which is differentiable end-to-end. This has been a topic of discussion since long, given patch based tokenization for images is largely unintuitive. While there have been earlier approaches to this, the authors claim that they present the first differentiable tokenizer that tokenizes according to semantics of the image rather than adhering to a grid based structure. The authors present their method which involves partitioning of tokens for different heirarchies, vertex (token) merging and pruning, and feature extraction (in a differentiable way). The authors present strong results on image classification, semantic segmentation, and image vectorization.

**Questions:**

Please see the strengths and weaknesses section

**Ethical Concerns:**

["NO or VERY MINOR ethics concerns only"]

**Final Justification:**

The paper introduces a strong, heirarchical tokenization approach which is well supported with experiments. The rebuttal also addressed all of my (mostly minor) concerns, and therefore I will continue to hold my Accept rating

**Limitations:**

yes

**Quality:**

4

**Strengths And Weaknesses:**

Strengths:

This is a refreshing take on tokenization for transformer based image processing, with a neatly defined intuitive method and strong results. The paper makes a lot of engineering choices and explains them well via ablations and theoretical motivations. The clever straight-through estimation trick enables e2e differentiation which makes the tokenizer really clean to train.

Weaknesses and Questions:

1. Currently, the merit of dHT for low resolution images, and subsequently tasks like image superresolution that rely on good abstraction of low res images is unclear. In Fig 6, authors mention that the default patch based tokenizer does better than their method for low resolution images as the grid based tokenization aligns well with semantic features. But shouldn't dHT also do the same? If not, does it mean that it becomes difficult to train dHT for lower resolution images? In that case, what does it mean for tasks like image superresolution which involves constructing a high res image from low res image(s)?
2. Latency numbers are missing in the paper. Having a tokenizer that works on multiple heirarchies can be costly. Can the authors present the throughputs of their and the default ViT methods at various resolutions? I understand tokenization might be different for different images, in which case the authors can consider reporting avg number of tokens and avg throughput over a stream of images for their method. Additionally, it is currently unclear how this latency requirement would scale with image size, it would be useful to understand that.
3. The core tokenization process, including the feature extractor and steps like mean injection, while effective in its current form, could be potentially simplified, or the steps can be swapped for something more end-to-end. While the question whether this leads to superior quality remains open, I believe it would be a direction worth exploring.

---

> ### Author Rebuttal · Authors · 2025-07-30
>
> ## Response to rgEH
>
> We thank the reviewer for all comments, suggestions, and contributions, which have served to improve the overall quality of our manuscript.
>
> ---
>
> ### Strengths
>
> > *authors present strong results on image classification, semantic segmentation, and image vectorization*
>
> > *refreshing take on tokenization for transformer based image processing, with a neatly defined intuitive method and strong results*
>
> > *makes engineering choices and explains them well via ablations and theoretical motivations*
>
> > *clever straight-through estimation trick enables e2e differentiation*
>
> We are grateful for the reviewer's remarks on the novelty of our motivation, and appreciate the comments on the central technical contributions in our work.
>
> ---
>
> ### Weaknesses and Questions
>
> > *the merit of dHT for low resolution images, and subsequently tasks like image super resolution that rely on good abstraction of low res images is unclear*
>
> We understand the reviewers concern. For very low resolutions (e.g. CIFAR's $32 \times 32$) a few of pixels can end up covering an entire region. In these settings, dHT will yield block-like regions similar to a square-patch tokenizer, since there are no clear inter-pixel edges to delineate,.
>
> However, we note that real world SR tasks are rarely defined at such low resolutions.
> - DIV2k: 4x task upscales from $510 \times 270$ to $2040 \times 1080$.
> - Urban100: 4x task typically upscales from $320 \times 320$ to $1280 \times 1280$.
>
> In these regimes, dHT performs well, and typically benefits from fewer overall tokens than square patch tokens with similar performance. Previous work on super resolution using superpixels (e.g., SPIN [[Zhang et al. 2023](https://openaccess.thecvf.com/content/ICCV2023/papers/Zhang_Lightweight_Image_Super-Resolution_with_Superpixel_Token_Interaction_ICCV_2023_paper.pdf)]) show strong results with reduced compute.
>
> We note that at *moderately* low resolutions, dHT generally performs quite well out-of-the-box compared to patch-based ViTs. To demonstrate this we include a simple small-scale test where we compare patch tokenization (DEITv3) to dHT on ViT-B16 on lower resolutions for ImageNet-val.
>
>
> | Tokenizer| $64 \times 64$ | $96 \times 96$  |
> | -------- | -------------- | --------------- |
> | Patch    | 26.01 (16 tok.)| 60.39 (36 tok.) |
> | dHT      | 59.64 (39 tok.)| 70.09 (58 tok.) |
>
>
> The result of this small scale experiment shows that dHT adapts better to lower resolution images, albeit with a higher token count due to adaptive tokenization. We emphasize that this is out-of-the-box with no fine-tuning, i.e. the ViT interpolates the learnable positional embedding dynamically, whereas dHT’skernelized positional embeddings doesn't require any adjustments.
>
> We thank the reviewer for their comment, and we will include this point in the camera ready version.
>
> ---
>
> > *Latency numbers are missing in the paper (...) tokenization might be different for different images, in which case the authors can consider reporting avg number of tokens and avg throughput over a stream of images*
>
> We thank the reviewer for their comment, and agree that latency numbers should be reported. We will include this in the camera ready version. As the reviewer points out, token counts can indeed be different between images.  Following your suggestion, we report *tokens-per-second* for dHT compared to baseline ViTs with base capacity, along with average tokens per image.
>
> | Capacity| Resolution| Tokenizer | Tok./s. | Tok./im.|
> |---------|-----------|-----------|---------|---------|
> | S16     | 224       | Patch     | 192.3k  | 197.0   |
> | S16     | 224       | dHT       |  94.2k  | 240.3   |
> | B16     | 224       | Patch     |  89.7k  | 197.0   |
> | B16     | 224       | dHT       |  68.2k  | 245.7   |
> | S16     | 384       | Patch     | 112.4k  | 577.0   |
> | S16     | 384       | dHT       |  58.1k  | 494.2   |
> | B16     | 384       | Patch     |  61.6k  | 577.0   |
> | B16     | 384       | dHT       |  48.9k  | 431.8   |
>
>
> *Throughput is computed over ImageNet-val (50k images), averaged over 4$\times$MI250x* with `fp32` precision.
>
> The results show that dHT incurs a moderate reduction in tokens processed per second compared to the standard patch tokenizer, as we discuss in our limitations section [L303]. However, as mentioned in our paper, the reduce in throughput is significantly lower at higher capacities and resolutions, which shows promise for high resolution tasks and video.
>
> ---
>
> > *The core tokenization process (...) could be potentially simplified (...) it would be a direction worth exploring.*
>
> We acknowledge that the design has potential for improvement, and mention this as future work and limitations in Sec.4.1 [L309]:
>
> *Our differentiable feature extraction method adheres to modularity and backward compatibility with ViTs. Although this provides unique advantages, the features are not necessarily optimal for adaptive regions. Further work is required on optimal feature extraction for superpixel tokens while maintaining fully differentiable partitioning mechanisms*
>
> In this work, our design was guided by our focus on retrofitting, and modeling decisions were made towards standard ViT commensurability to directly allow backward compatability with existing models. If one were willing to forgo this requirement, a more parsemoneous implementation could indeed be possible, but at the expense of backward compatability, which we claim as one of our central contributions. However, we agree with reviewer's sentiment, this is a clearly a direction worth exploring in future work.

---

> > ### Comment · Reviewer_rgEH · 2025-08-07
> >
> > I thank the authors for their detailed rebuttal and will continue to hold my positive assessment of the paper. Thanks for the good work!

---

> > > ### Author Response · Authors · 2025-08-07
> > >
> > > We appreciate the encouragement and thank the reviewer for confirming that they are maintaining their positive evaluation.

---

### Official Review · Reviewer_m47p · 2025-07-03

**Clarity:** 3
**Significance:** 4
**Originality:** 4
**Rating:** 5
**Confidence:** 4

**Summary:**

The author presents dHT, namely differentiable hierarchical tokenization, aiming to address the problem in vision tokenization that text tokenizers align with semantic units, yet patch-based vision tokenizers fragment objects without regard for their structure. dHT provides a tokenization with pixel-level granularity by constructing an optimal partition via hierarchical vertex merging. While the tokenization’s multiscale adaptability and redundancy management are achieved through pruning using the information criterion. dHT also achieves differentiability through weighted aggregation and a novel mean-injection trick. Moreover, dHT can retrofit any vision transformer without altering its layers or positional embedding shape. Experiments show that dHT has competitive performance in both image-level classification and dense-prediction tasks, also supports out-of-the-box raster-to-vector conversion. An extensive ablation study proves the effectiveness of each proposed module towards the whole network.

**Questions:**

Instead of the most similar neighbor in image coordinate space, could the merging process be based on the most similar neighbor in feature space so that the resulting image patch might not necessarily be continuous? In other words, could the entire graph be built on feature space?

As described in the introduction section, the main purpose of this work is to make the image tokenization similar to the text token by precise semantic alignment. Will the authors try such an image tokenization method on VLM, where text comes together with an image? Will dHT perform better in aligning image and text tokens?

**Ethical Concerns:**

["NO or VERY MINOR ethics concerns only"]

**Final Justification:**

The authors addressed my concerns. I recommend accept.

**Limitations:**

See weakness and the question part.

**Quality:**

4

**Strengths And Weaknesses:**

**Strength:**
Mainly based on graph theory, the author provides strong theoretical support for every design in the network architecture.

dHT treats the searching for optimal partition as a model selection problem using information criteria offers a clear objective without the need for threshold calibration.

The author uses a novel mean-injection trick that provides a clever and easy way to make the generalized ViT features fully differentiable
dHT is a plug‑and‑play visual tokenizer that can retrofit any vision transformer, offering high modularity for reuse and backward compatibility

**Weakness:**
Line 158 typo ‘sytstems’  -> systems

In dHT, while selecting tokens is made differentiable, the merging process remains handcrafted. A learnable merging process might result in further gains in performance.

The extra cost for constructing such a deformable image token should also be reported compared with fixed patch methods in terms of inference efficiency.

---

> ### Author Rebuttal · Authors · 2025-07-30
>
> ## Response to m47p
>
> We thank the reviewer for insightful comments and insights, and for their time invested in reviewing our work.
>
> ---
>
> ### Strengths
>
> > *the author provides strong theoretical support for every design in the network architecture*
>
> > *information criteria offers a clear objective without the need for threshold calibration*
>
> > *mean-injection trick that provides a clever and easy way to make the generalized ViT features fully differentiable*
>
> We appreciate the reviewers remarks on the theoretical foundation of our work, as well as the novelty of our contributions to the field.
>
> ---
>
> ### Weaknesses
>
> > *Line 158 typo ‘sytstems’ -> systems*
>
> We thank the reviewer for pointing out the typo we missed at the submission of our manuscript. We will correct this, and perform additional proofreading for the camera ready version.
>
> ---
>
> > *The extra cost (...) should also be reported compared with fixed patch methods in terms of inference efficiency*
>
> We agree that reporting computational overhead is important. We will add a throughput analysis to the camera‑ready version.
>
> Given the adaptive of dHT, token counts are different depending on the image. Following reviewer rgEH's suggestion, we report *tokens-per-second* for dHT compared to baseline ViTs with base capacity, along with average tokens per image. This provides balanced evaluation of the computational overhead associated with dHT.
>
> | Capacity| Resolution| Tokenizer | Tok./s. | Tok./im.|
> |---------|-----------|-----------|---------|---------|
> | S16     | 224       | Patch     | 192.3k  | 197.0   |
> | S16     | 224       | dHT       |  94.2k  | 240.3   |
> | B16     | 224       | Patch     |  89.7k  | 197.0   |
> | B16     | 224       | dHT       |  68.2k  | 245.7   |
> | S16     | 384       | Patch     | 112.4k  | 577.0   |
> | S16     | 384       | dHT       |  58.1k  | 494.2   |
> | B16     | 384       | Patch     |  61.6k  | 577.0   |
> | B16     | 384       | dHT       |  48.9k  | 431.8   |
>
>
> *Throughput is computed over ImageNet-val (50k images), averaged over 4$\times$MI250x*.
>
> The results show that dHT incurs a moderate reduction in tokens processed per second compared to the standard patch tokenizer, as we discuss in our limitations section [L303]. However, as mentioned in our paper, the reduce in throughput is significantly lower at higher capacities and resolutions, which shows promise for high resolution tasks and video.
>
> ---
>
> ### Questions
>
> > *Instead of the most similar neighbor in image coordinate space, could the merging process be based on the most similar neighbor in feature space so that the resulting image patch might not necessarily be continuous?*
>
> Exploring non-continuous extensions to dHT is an intriguing proposal and a promising direction for future work. However, our current framework would need substantial modifications before such an approach could be made viable.
>
> - Our construction guarantees that every region remains a single connected component in the grid graph, which is a condition for the theoretical results in Appendix A (Prop.A.13) and the degrees of freedom estimate (Appendix B, Eq.B.5) by reciprocal graph volume. These are central for IC pruning. If we allow non-planar edges, these results no longer hold, so an alternative IC penalty term would need to be derived.
> - The mean-injection trick (Sec.2.2, Eq.6) via interpolated feature extraction requires regions to be reasonably dense. If a region is scattered over the image, the resulting mask becomes exceedingly sparse, which could potentially lead to vanishing gradients in end-to-end training. Alternative feature extraction frameworks could be explored, but this would make modularity and retrofitting more challenging.
> - Allowing non-contiguous patches increases the search space for merging from $O(N) \to O(N^2)$ for $N=HW$, which makes early merges significantly less tractable. Tree-based partitioning could provide an interesting approach, but would require a fundamentally different approach than dHT.
>
> This does not prevent a global disjoint merging strategy in the tokenizer, but key components of our proposed framework would have to be redesigned to accommodate such an approach. Alternatively, non-local merging within the ViT (e.g. adapting Bolya et al. [22]) could be combined with dHT without altering the tokenizer itself, but lies beyond the scope of this submission.
>
> We thank the reviewer for the suggestion and will add the discussion on non-contiguous tokenisation as a promising avenue for further work in the revised manuscript.
>
> ---
>
> > *the main purpose of this work is to make the image tokenization similar to the text token by precise semantic alignment. Will the authors try such an image tokenization method on VLM, where text comes together with an image? Will dHT perform better in aligning image and text tokens?*
>
> The extension to VLMs is indeed a research direction we are pursuing. Our preliminary investigations have centered around OCR and document-focused VLMs (docVLMs) [[Faysse et al. 2025](https://arxiv.org/abs/2407.01449), [Souibgui et al. 2025](https://arxiv.org/abs/2505.07496)], which has become a standard pre-training objective for modern VLMs. BLIP‑2 [[Li et al. 2023](https://arxiv.org/abs/2301.12597)], LLaVA-NeXT [[Liu et al. 2024](https://arxiv.org/abs/2407.07895)], and CogVLM [[Wang et al. 2024](https://arxiv.org/abs/2311.03079)] all pre‑train on DocVQA‑style corpora alongside natural image captions.
>
> Adaptive tokenization can effectively address key limitations of current docVLM frameworks. We observe that square-patch tokens often become a bottleneck, as these are poorly aligned with the more heterogeneous structure inherent in scanned documents. Preliminary findings suggest that dHT can help resolve this by generating adaptive tokens that better align with the semantics on a per-sample basis.
>
> More generally, we expect adaptive tokenization to be beneficial for general VLMs as well, but our initial investigations focus on the docVLMs, since this is becoming a canonical pretext task. While we do not present experiments on VLMs in this submission, we believe adaptive tokenization is well suited to VLM tasks, and we are in progress with further development on this line of work.

---

> > ### Comment · Reviewer_m47p · 2025-08-09
> > **Addressed my concerns**
> >
> > Thanks for the rebuttal. The authors addressed most of my concerns, I maintain the score.

---

> > > ### Author Response · Authors · 2025-08-09
> > >
> > > Thank you for letting us know that our response resolved most of your concerns, and for reaffirming the positive assessment of our work.

---

### Author Response · Authors · 2025-08-07

Many thanks to all reviewers for their thoughtful engagement and positive feedback. We look forward to integrating your suggestions and improvements into the revised manuscript.

---

### Decision · Program_Chairs · 2025-09-17

**Decision:**

Accept (spotlight)

**Comment:**

The authors proposed an algorithm for differentiable image tokenization. All the reviewers agree the novelty of the paper. The experiments are well and the paper is well written. The problem they tackled is important and fundamental. I believe it will be an impactful work.